# SPOT: Scalable 3D Pre-training via Occupancy Prediction for Autonomous Driving

## Abstract

Annotating 3D LiDAR point clouds for perception tasks including 3D object detection and LiDAR semantic segmentation is notoriously time-and-energy-consuming. To alleviate the burden from labeling, it is promising to perform large-scale pre-training and fine-tune the pre-trained backbone on different downstream datasets as well as tasks. In this paper, we propose SPOT, namely **S**calable **P**re-training via **O**ccupancy prediction for learning **T**ransferable 3D representations, and demonstrate its effectiveness on various public datasets with different downstream tasks under the label-efficiency setting. Our contributions are threefold: (1) Occupancy prediction is shown to be promising for learning general representations, which is demonstrated by extensive experiments on plenty of datasets and tasks. (2) SPOT uses beam re-sampling technique for point cloud augmentation and applies class-balancing strategies to overcome the domain gap brought by various LiDAR sensors and annotation strategies in different datasets. (3) Scalable pre-training is observed, that is, the downstream performance across all the experiments gets better with more pre-training data. We believe that our findings can facilitate understanding of LiDAR point clouds and pave the way for future exploration in LiDAR pre-training. Codes and models will be released.

## 1 Introduction

Light Detection And Ranging (LiDAR), which emits and receives laser beams to accurately estimate the distance between the sensor and objects, serves as one of the important sensors in outdoor scenes, especially for autonomous driving. The return of LiDAR is a set of points in the 3D space, each of which contains location (the XYZ coordinates) and other information like intensity and elongation. Taking these points as inputs, 3D perception tasks like 3D object detection and semantic segmentation aim to predict 3D bounding boxes or per-point labels for different objects including cars, pedestrians, cyclists, and so on, which are prerequisites for downstream safety control tasks.

In the past few years, research on learning-based 3D perception methods flourishes (Yan et al., 2018; Yin et al., 2021; Shi et al., 2020; 2023; Zhu et al., 2021; Zhang et al., 2023) and achieves unprecedented performance on different published datasets (Geiger et al., 2012; Behley et al., 2019; Mao et al., 2021; Caesar et al., 2020; Sun et al., 2020). However, these learning-based methods are data-hungry and it is notoriously time-and-energy-consuming to label 3D point clouds. On the contrary, large-scale pre-training and fine-tuning with fewer labels in downstream tasks serves as a promising solution to improve the performance in label-efficiency setting. Previous methods can be divided into two streams: (1) Embraced by AD-PT (Yuan et al., 2023), semi-supervised pre-training achieves a strong performance gain when using fewer labels but limited to specific task like 3D object detection (**task-level** gap). (2) Other works including GCC-3D (Liang et al., 2021), STRL (Huang et al., 2021), BEV-MAE (Lin & Wang, 2022), CO3 (Chen et al., 2022) and MV-JAR (Xu et al., 2023) utilize unlabeled data for pre-training. This branch of work fails to generalize across datasets with different LiDAR sensors and annotation strategies, as shown in Fig. 1b. (**dataset-level** gap)

To overcome both **task-level** and **dataset-level** gaps and learn general representations, we propose SPOT, namely **S**calable **P**re-training via **O**ccupancy prediction for learning **T**ransferable representation. Firstly, we argue that occupancy prediction serves as a more general pre-training task for task-level generalization, as compared to 3D object detection and LiDAR semantic segmentation. The reason lies in that occupancy prediction is based on denser voxel-level labels with abundant

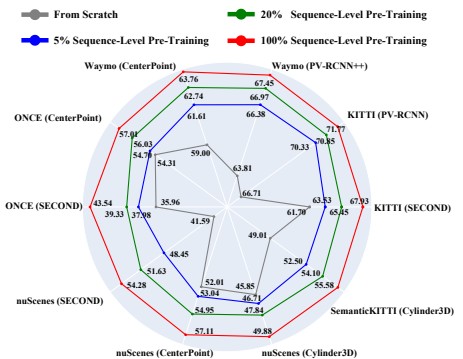 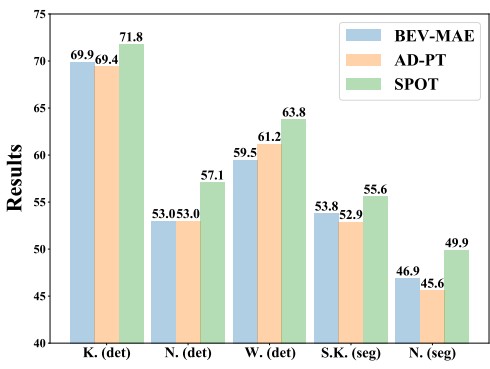

(a) Scalability across various datasets and tasks.  (b) Comparison with other pre-training methods.

Figure 1: (a) SPOT pre-trains the 3D and 2D backbones and achieves scalable performance improvement across various datasets and tasks in label-efficient setting. Different colors indicate different amounts of pre-training data. (b) SPOT delivers the best performance on various datasets and tasks among different pre-training methods. " K. (det) ", " N. (det) ", " W. (det) " are abbreviations for KITTI, nuScenes, and Waymo detection tasks, while " S.K. (seg) " and " N. (seg) " are abbreviations for SemanticKITTI, and nuScenes segmentation tasks, respectively.

classes, which incorporates spatial information similar to 3D object detection as well as semantic information introduced in semantic segmentation. Secondly, as the existing datasets use LiDAR sensors with various numbers of laser beams and different category annotation strategies, we propose to use beam re-sampling for point cloud augmentation and class-balancing strategies to overcome these domain gaps. Beam re-sampling augmentation simulates LiDAR sensors with different numbers of laser beams to augment point clouds from a single source pre-training dataset, alleviating the domain gap brought by LiDAR types. Class-balancing strategies apply balance sampling on the dataset and category-specific weights on the loss functions to narrow down the annotation gap. Last but not least, we observe that more pre-training data bring better downstream performance towards different tasks. This indicates that SPOT is a scalable pre-training method for LiDAR point clouds, which paves the way for future large-scale 3D representation learning in autonomous driving.

Our contributions can be summarized into three aspects: (1) SPOT demonstrates that occupancy prediction is a promising pre-training method for general and scalable 3D representation learning on LiDAR point cloud. (2) Beam re-sampling augmentation and class-balancing strategies are useful in narrowing domain gaps introduced by different LiDAR sensors and annotation strategies. (3) Extensive experiments are conducted on different 3D perception tasks and various datasets including Waymo (Sun et al., 2020), nuScenes (Caesar et al., 2020), ONCE (Mao et al., 2021), KITTI (Geiger et al., 2012), and SemanticKITTI (Behley et al., 2019) to demonstrate the effectiveness of SPOT. As shown in Fig. 1, SPOT (a) continuously improves the downstream performance as more pre-training data are used, and (b) learns general representations and brings more consistent improvement as compared to previous pre-training methods.

## 2 RELATED WORK

**LiDAR 3D Perception.** There are two main tasks on LiDAR point clouds: 3D object detection and LiDAR semantic segmentation, both of which are essential for scene understanding and control tasks. Current LiDAR 3D detectors can be divided into three main classes based on the 3D backbone in the architectures. (1) Point-based 3D detector embeds point-level features to predict 3D bounding boxes, which is embraced by PointRCNN (Shi et al., 2019). (2) Voxel-based 3D detector divides the surrounding environment of the autonomous vehicle into 3D voxels and uses sparse convolution or transformer-based encoder to generate voxel-level features for detection heads. Second (Yan et al., 2018) and CenterPoint (Yin et al., 2021) are popular and SOTA voxel-based 3D detectors. (3) Point-and-voxel-combined method like PV-RCNN (Shi et al., 2020) and PV-RCNN++ (Shi et al., 2023) utilize both voxel-level and point-level features. For LiDAR semantic segmentation task, the goal is to predict a category label for each point in the LiDAR point clouds. Cylinder3D (Zhu et al., 2021),

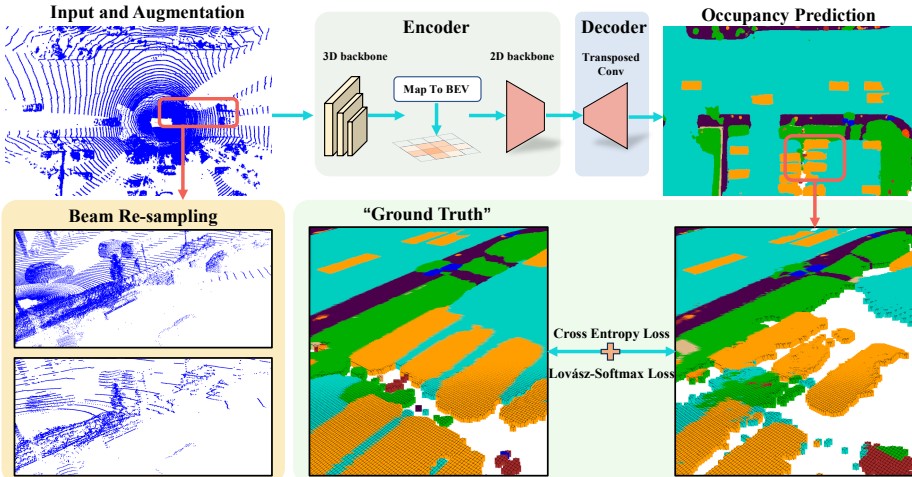

Figure 2: The overview of the proposed SPOT. Firstly, the input LiDAR point cloud is augmented by beam re-sampling to simulate various LiDAR sensors, which helps learn general representations. Then point clouds are processed by backbone encoders consisting of 3D and 2D ones, which are utilized to initialize downstream architectures after pre-training. Next, a lightweight decoder with stacked transposed convolutions embeds the BEV features to further predict occupancy probability. Finally, we use class-balancing cross entropy loss and Lovász-Softmax loss to guide the pre-training.

the pioneering work on this task, proposes to first apply the 3D backbone to embed the voxel-level features and then a decoder for final semantic label predictions. All these methods are data-hungry and labeling for 3D point clouds is time-and-energy-consuming. **To reduce the labeling burden, previous works explore semi-supervised learning (Unal et al., 2022; Kong et al., 2023; Li et al., 2023a) and achieve excellent performance, but they are limited to specific task. In this work, we explore general 3D representation learning via large-scale pre-training.**

**Large-scale Pre-training for Label-efficient Learning in LiDAR 3D Perception.** It is promising to reduce labeling burdens by large-scale pre-training. There are two branches of methods. The first one, embraced by AD-PT (Yuan et al., 2023), is semi-supervised pre-training for 3D detection on LiDAR point cloud. AD-PT demonstrates a strong performance gain when using fewer labels. However, it suffers from the limited downstream tasks (3D object detection only). The second branch of methods include GCC-3D (Liang et al., 2021), STRL (Huang et al., 2021), CO3 (Chen et al., 2022), OCC-MAE (Min et al., 2023), BEV-MAE (Lin & Wang, 2022) and MV-JAR (Xu et al., 2023), which utilize unlabeled data for pre-training. But these methods fails to generalize across different LiDAR sensors. In this work, we propose SPOT to pre-train the 3D backbone for LiDAR point clouds and improve performance in different downstream tasks with various sensors and architectures, as shown in Fig. 1.

**Semantic Occupancy Prediction.** The primary objective is to predict whether a voxel in 3D space is free or occupied as well as the semantic labels for the occupied ones, which enables a comprehensive and detailed understanding of the 3D environment. Represented by MonoScene (Cao & de Charette, 2022), VoxFormer (Li et al., 2023b), TPVFormer (Huang et al., 2023), JS3C-Net (Yan et al., 2021) and SCPNet (Xia et al., 2023), deep learning methods achieve unprecedented performance gains on this task. However, these methods are specially designed for semantic occupancy prediction task and fail to learn general representations for different 3D perception tasks, such as object detection and semantic segmentation. In this paper, SPOT is proposed to use 3D semantic occupancy prediction to learn a unified 3D scene representation for various downstream tasks including 3D object detection and LiDAR semantic segmentation.

## 3 METHOD

We discuss the proposed SPOT in detail. As shown in Fig. 2, SPOT contains four parts: (a) Augmentations on LiDAR point clouds. (b) Encoder for LiDAR point clouds to generate BEV features,

which are pre-trained and used for different downstream architectures and tasks. (c) Decoder to predict occupancy based on BEV features. (d) Loss function with class-balancing strategy. We first introduce the problem formulation as well as the overall pipeline in Sec. 3.1. Then we respectively discuss beam re-sampling augmentation and class-balancing strategies in Sec. 3.2 and Sec. 3.3.

## 3.1 PROBLEM FORMULATION AND PIPELINE

**Notation.** To start with, we denote LiDAR point clouds $\mathbf{P} \in \mathbb{R}^{N \times (3+d)}$ as the concatenation of $xyz$-coordinate $\mathbf{C} \in \mathbb{R}^{N \times 3}$ and features for each point $\mathbf{F} \in \mathbb{R}^{N \times d}$, that is $\mathbf{P} = [\mathbf{C}, \mathbf{F}]$. $N$ here is the number of points and $d$ represents the number of point feature channels, which is normally $d = 1$ for intensity of raw input point clouds. Paired with each LiDAR point cloud, detection labels $L_{det} \in \mathbb{R}^{N_{det} \times 10}$ and segmentation labels for each point $L_{seg}^j \in \{0, 1, 2, ..., N_{\text{cls}}\}$ ($j = 1, 2, ..., N$) are provided. For detection labels, $N_{det}$ is the number of 3D boundary boxes in the corresponding LiDAR frame and each box is assigned $xyz$-location, sizes in $xyz$-axis (length, width and height), orientation in $xy$-plane (the yaw angle), velocity in $xy$-axis and the category label for the corresponding object. For segmentation labels, each LiDAR point is assigned a semantic label where 0 indicates "empty", and 1 to $N_{\text{cls}}$ are different categories like vehicle, pedestrian, and so on.

**Pre-processing.** We generate "ground-truth" occupancy $\mathbf{O} \in \{0, 1, 2, ..., N_{\text{cls}}\}^{H \times W}$ for pre-training following the practice in (Tian et al., 2023), where $H$ and $W$ are respectively number of voxels in $xy$-axis and Fig. 2 shows an example. In general, we take LiDAR point clouds in the same sequence along with their detection and segmentation labels as the inputs, and divide the labels into dynamic and static. After that, all LiDAR point clouds in that sequence can be fused to generate dense point clouds, followed by mesh reconstruction to fill up the holes. Finally, based on the meshes, we can obtain occupancy $\mathbf{O}$. For more details, please refer to (Tian et al., 2023).

**Encoding and Decoding.** Given an input LiDAR point cloud $\mathbf{P} \in \mathbb{R}^{N \times (3+d)}$, augmentations including beam re-sampling, random flip and rotation, are first applied and result in augmented point cloud $\mathbf{P}_{\text{aug}} \in \mathbb{R}^{N \times (3+d)}$. Then $\mathbf{P}_{\text{aug}}$ is embedded with sparse 3D convolution and BEV convolution backbones and obtain dense BEV features $\mathbf{F}_{\text{BEV}} \in \mathbb{R}^{\hat{H} \times \hat{W} \times \hat{d}}$ as follows:

$$\mathbf{F}_{\text{BEV}} = f^{\text{enc}}(\mathbf{P}_{\text{aug}}), \tag{1}$$

where $\hat{H}$ and $\hat{W}$ are height and width of the BEV feature map and $\hat{d}$ is the number of feature channels after encoding. Then based on $\mathbf{F}_{\text{BEV}}$, a convolution decoder together with a Softmax operation (on the last dimension) is applied to generate dense occupancy probability prediction $\hat{\mathbf{O}} \in \mathbb{R}^{H \times W \times (N_{\text{cls}}+1)}$ using the following equation:

$$\hat{\mathbf{O}} = \text{softmax}(f^{\text{dec}}(\mathbf{F}_{\text{BEV}})), \tag{2}$$

where $H$ and $W$ are the same as those of $\mathbf{O}$. For each pixel on BEV map, an $N_{\text{cls}} + 1$ dimensional probability vector is predicted, each entry of which indicates the probability of the corresponding category. The decoder $f^{\text{dec}}$ is kept simple and lightweight. It consists of only three layers of 2D transposed convolution with a kernel size of 3 and a prediction head composed of linear layers.

**Loss Function.** To guide the encoders to learn transferable representations, class-balancing cross-entropy loss and Lovász-Softmax loss (Berman et al., 2018) are applied on the predicted occupancy probability $\hat{\mathbf{O}}$ and the "ground-truth" occupancy $\mathbf{O}$. The overall loss can be written by:

$$\mathcal{L} = \mathcal{L}_{\text{ce}}(\mathbf{O}, \hat{\mathbf{O}}) + \lambda \cdot \mathcal{L}_{\text{lov}}(\mathbf{O}, \hat{\mathbf{O}}), \tag{3}$$

where $\lambda$ is the weighting coefficient used to balance the contributions of the two loss. For class-balancing cross-entropy loss, details are discussed in Sec. 3.3. And the Lovász-Softmax loss is a popular loss function used in semantic segmentation, whose formulation is as follows:

$$\mathcal{L}_{\text{lov}}(\mathbf{O}, \hat{\mathbf{O}}) = \frac{1}{N_{\text{cls}}} \sum_{n=1}^{N_{\text{cls}}} \overline{\Delta_{J_c}}(\mathbf{M}(n)), \quad \mathbf{M}(n)_{h,w} = \begin{cases} 1 - \hat{\mathbf{O}}_{h,w,n} & if \ n = \mathbf{O}_{h,w} \\ \hat{\mathbf{O}}_{h,w,n} & otherwise \end{cases}, \tag{4}$$

where $\mathbf{M}(n) \in \mathbb{R}^{H \times W}$ means the errors of each pixel on BEV map of class $n$, and $h, w$ is the pixel index for the BEV map. $\overline{\Delta_{J_c}}$ denotes the Lovász extension of the Jaccard index to maximize the Intersection-over-Union (IoU) score for class $n$, which smoothly extends the Jaccard index loss based on a submodular analysis of the set function (Berman et al., 2018).

### 3.2 BEAM RE-SAMPLING AUGMENTATION

Different datasets use different LiDAR sensors to collect data. The most significant coefficient that brings domain gap is the beam numbers of LiDAR sensors, which directly determines the sparsity of the return point clouds. Fig. 3 shows an example where two LiDAR point clouds are collected by different LiDAR sensors in the same scene and it can be found that 16-beam LiDAR brings a much sparser point cloud, which results in varying distributions of the same object and degrades the performance. In order to learn general representations that benefit various datasets, we propose equivalent LiDAR beam sampling to diversify the pre-training data.

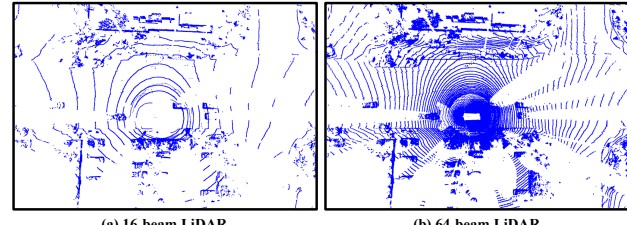

(a) 16-beam LiDAR            (b) 64-beam LiDAR

Figure 3: Examples of different LiDAR beams.

First of all, we quantify the sparsity of point clouds collected by different LiDAR sensors. The dominant factor is beam-number and the Vertical Field Of View (VFOV) also matters. We calculate the beam density by the following Eq. 5, where $N_{\text{beam}}$ is the number of the LiDAR beam, and $\alpha_{\text{up}}$ and $\alpha_{\text{low}}$ respectively represent the upper and lower limits of the vertical field of view of the sensor,

$$B_{\text{density}} = \frac{N_{\text{beam}}}{\alpha_{\text{up}} - \alpha_{\text{low}}}. \tag{5}$$

Next, by dividing $B_{\text{density}}$ of different downstream datasets with that of the pre-training dataset, we compute re-sampling factors $R_{\text{sample}}$. Re-sampling is conducted for the pre-training data according to different $R_{\text{sample}}$. Specifically, given the original LiDAR point cloud, we transform the Cartesian coordinates $(x, y, z)$ of each point into the spherical coordinates $(r, \phi, \theta)$, where $(r, \phi, \theta)$ are the range, inclination and azimuth, respectively. Finally, uniform re-sampling is conducted on the dimension of inclination. The transformation function can be formulated by:

$$r = \sqrt{x^2 + y^2 + z^2}, \ \ \phi = arctan(x/y), \ \ \theta = arctan(z/\sqrt{x^2 + y^2}). \tag{6}$$

### 3.3 CLASS-BALANCING STRATEGIES

The contribution to downstream tasks of different categories varies. First, different datasets have various distributions over categories, which causes domain gaps and hinders learning general representations. Also, in 3D detection task, foreground classes like vehicle, pedestrian and cyclist are more important than background categories including pavement and vegetation. Thus, we propose class-balancing strategies respectively on the dataset and loss function to narrow the domain gaps.

**Dataset Balancing.** Considering that background classes are almost ubiquitous in every scene, we focus solely on the foreground classes in the dataset, such as cars, pedestrians, cyclists and so on. As shown in Fig. 4, we conducted a statistical analysis of the distribution of foreground semantic classes in the pre-training dataset, and it is evident that the pre-training dataset has a severe class imbalance problem. Inspired by (Zhu et al., 2019), (Zou et al., 2018) and (Unal et al., 2022), we em-

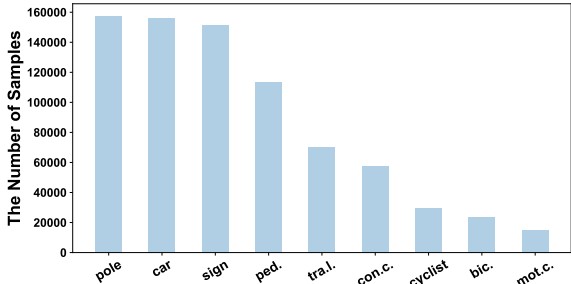

Figure 4: Distribution of different classes.

ploy a frame-level re-sampling strategy to alleviate the severe class imbalance. Assuming that there are $N_{\text{fg}}$ foreground classes, we calculate the class sampling weights $s_i$ ($i = 1, 2, ..., N_{\text{fg}}$) for each class based on the proportion of samples:

$$s_i = \sqrt{m/n_i}, \ \ m = \frac{1}{N_{\text{fg}}}, \ \ n_i = \frac{N_i}{\sum_{j=1}^{N_{\text{fg}}} N_j}, \tag{7}$$

where $N_i$ is the number of samples for the $i^{th}$ class. Fewer samples in a category brings higher weight $s_i$ for it. Based on the sampling weights, we can employ a random duplication to balance

the classes and compose the final dataset to alleviate the class imbalance. This is advantageous as it allows us to learn scene representations more effectively in the pre-training task, facilitating downstream tasks.

**Loss Function Balancing.** In real-world scenarios, the surrounding 3D space of the autonomous vehicle is dominated by unoccupied states or background information. This can be harmful to the training process because the loss would be overwhelmed by a substantial amount of useless information. To overcome this challenge, we propose to assign different weights to different categories. Specifically, we assign weight $w_{\text{fg}} = 2.0$ to common foreground categories including car, pedestrian, cyclist, bicycle, and motorcycle. Meanwhile, other background categories like vegetation and road are assigned $w_{\text{bg}} = 1.0$ and $w_{\text{empty}} = 0.01$ for unoccupied voxels.

# 4 EXPERIMENTS

The goal of pre-training is to learn general representations for various downstream tasks, datasets, and architectures. In this section, we design extensive experiments to answer the question whether SPOT learns such representations in a label-efficiency way. We first introduce experiment setup in Sec. 4.1, followed by main results with baselines in Sec. 4.2. Then we also provide discussions about pre-training tasks selection, ablation study and performance on full downstream datasets in Sec. 4.3. Finally, we end this section with visualization about 3D object detection results.

## 4.1 EXPERIMENTAL SETUP

**Pre-training Dataset.** We use the *Waymo Open dataset* (Sun et al., 2020) as our pre-training dataset, which uses a main 64-beam LiDAR and 4 short-range LiDARs to collect point clouds. Waymo contains 798 sequences and 202 sequences for training and validation, respectively. Following the methodology mentioned in Sec. 3.1, we generate dense occupancy labels for each sample where $N_{\text{cls}} = 15$. This means 15 semantic categories including car, pedestrian and motorcycle, as well as "empty" are marked for each voxel. To evaluate the scalability of SPOT, we partition Waymo into 5%, 20%, and 100% subsets at the sequence level and perform the pre-training on different subsets.

**Downstream Tasks.** Popular LiDAR perception tasks include 3D object detection and LiDAR semantic segmentation. For detection, we cover the vast majority of currently available datasets, including *KITTI* (Geiger et al., 2012), *NuScenes* (Caesar et al., 2020) and *ONCE* (Mao et al., 2021) with popular 3D detectors including SECOND (Yan et al., 2018), CenterPoint (Yin et al., 2021) and PV-RCNN (Shi et al., 2020) for evaluation. *NuScenes* utilizes a 32-beam LiDAR to collect 40,000 LiDAR point clouds, of which 28,130 samples are used for training and 6,019 samples for validation. We evaluate the performance using the official Mean Average Precision (mAP) and NuScenes Detection Score (NDS) (Caesar et al., 2020). *KITTI* consists of 7,481 samples for training and 7,518 samples for validation collected with a 64-beam LiDAR. We report the results using three levels of mAP metrics: easy, moderate, and hard, following the official settings in (Geiger et al., 2012). *ONCE* contains 19k labeled LiDAR point clouds, of which 5k point clouds are used for training, 3k for validation and 8k for testing, all of which are collected by a 40-beam LiDAR. For evaluation, we follow (Mao et al., 2021) to use the mAP metrics by different ranges: 0-30m, 30-50m, and 50m-Inf. For semantic segmentation, we conduct experiments on *SemanticKITTI* (Behley et al., 2019) and *NuScenes* (Caesar et al., 2020) with the famous LiDAR segmentor Cylinder3D (Zhu et al., 2021). *SemanticKITTI* has 22 point cloud sequences and is divided into a train set with 19,130 samples together with a validation set with 4,071 frames. The evaluation metric of the two datasets adopts the commonly used mIoU (mean Intersection over Union). To compute mIoU, per-category IoU is first computed as $\text{IoU}_i = \frac{\text{TP}_i}{\text{TP}_i + \text{FP}_i + \text{FN}_i}$, where $\text{TP}_i$, $\text{FP}_i$ and $\text{FN}_i$ denote true positive, false positive and false negative for class $i$, respectively. Then IoUs for different classes are averaged to get the final mIoU.

**Baseline Methods.** We select two representative pre-training methods for unsupervised (BEV-MAE (Lin & Wang, 2022)) and supervised (AD-PT (Yuan et al., 2023)) branches respectively.

**Implementation Details.** For pre-training phase, we adopt commonly used 3D and 2D backbones in (Yan et al., 2018; Yin et al., 2021; Shi et al., 2020) and $N_{\text{cls}} = 15$, $\lambda = 1$. We train 30 epochs with the Adam optimizer, using the one-cycle policy with a learning rate of 0.003. For the downstream detection task, we train 30 epochs for NuScenes, 80 epochs for KITTI and ONCE. For the down-

Table 1: Fine-tuning performance on NuScenes benchmark. P.D.A. represents the Pre-training Data Amount. We fine-tune on 5% NuScenes training data.

| Detector | Method | P.D.A. | mAP | NDS | Car | Truck | CV. | Bus | Trailer | Barrier | Motor. | Bicycle | Ped. | TC. |
|---|---|---|---|---|---|---|---|---|---|---|---|---|---|---|
| SECOND | From Scratch | - | 32.16 | 41.59 | 69.13 | 33.94 | 10.12 | 46.56 | 17.97 | 32.34 | 15.87 | 0.00 | 57.30 | 37.99 |
| | BEV-MAE (Lin & Wang, 2022) | 100% | 32.09 | 42.88 | 69.84 | 34.79 | 8.19 | 48.36 | 22.46 | 32.67 | 13.01 | 0.13 | 56.10 | 35.33 |
| | AD-PT (Yuan et al., 2023) | 100% | 37.69 | 47.95 | 74.89 | 41.82 | 12.05 | 54.77 | 28.91 | 34.41 | 23.63 | 3.19 | 63.61 | 39.54 |
| | SPOT (ours) | 5% | 37.96 | 48.45 | 74.74 | 37.94 | 12.17 | 54.94 | 27.69 | 38.03 | 22.91 | 2.55 | 64.27 | 44.31 |
| | SPOT (ours) | 20% | 39.63 | 51.63 | 75.58 | 41.41 | 12.95 | 55.67 | **29.92** | 40.13 | 23.26 | 4.77 | 70.40 | 42.18 |
| | SPOT (ours) | 100% | **42.57** | **54.28** | **76.98** | **42.86** | **14.54** | **59.56** | 29.30 | **44.04** | **30.91** | 7.52 | **72.70** | **47.26** |
| CenterPoint | From Scratch | - | 42.37 | 52.01 | 77.13 | 38.18 | 10.50 | 55.87 | 23.43 | 50.50 | 35.13 | 15.18 | 71.58 | 46.16 |
| | BEV-MAE (Lin & Wang, 2022) | 100% | 42.86 | 52.95 | 77.35 | 39.95 | 10.87 | 54.43 | 25.03 | 51.20 | 34.88 | 15.15 | 72.74 | 46.96 |
| | AD-PT (Yuan et al., 2023) | 100% | 44.99 | 52.99 | 78.90 | **43.82** | 11.13 | 55.16 | 21.22 | **55.10** | 39.03 | 17.76 | 72.28 | **55.43** |
| | SPOT (ours) | 5% | 43.56 | 53.04 | 77.21 | 38.13 | 10.45 | 56.41 | 24.19 | 50.33 | 37.74 | 18.55 | 73.97 | 48.59 |
| | SPOT (ours) | 20% | 44.94 | 54.95 | 78.30 | 40.49 | 12.32 | 56.68 | 28.10 | 51.77 | 35.93 | 22.46 | 75.98 | 47.38 |
| | SPOT (ours) | 100% | **47.47** | **57.11** | **79.01** | 42.41 | **13.04** | 59.51 | 29.53 | 54.74 | **42.54** | **24.66** | **77.65** | 51.65 |

Table 2: Fine-tuning performance (AP$_{3D}$) on KITTI benchmark. P.D.A. represents the Pre-training Data Amount, and fine-tuning is performed on 20% KITTI training data.

| Detector | Method | P.D.A. | mAP | Car | | | Pedestrian | | | Cyclist | | |
|---|---|---|---|---|---|---|---|---|---|---|---|---|
| | | | (Mod.) | Easy | Mod. | Hard | Easy | Mod. | Hard | Easy | Mod. | Hard |
| SECOND | From Scratch | - | 61.70 | 89.78 | 78.83 | 76.21 | 52.08 | 47.23 | 43.37 | 76.35 | 59.06 | 55.24 |
| | BEV-MAE (Lin & Wang, 2022) | 100% | 63.45 | 89.50 | 78.53 | 75.87 | 53.59 | 48.71 | 44.20 | 80.73 | 63.12 | 58.96 |
| | AD-PT (Yuan et al., 2023) | 100% | 65.95 | 90.23 | 80.70 | **78.29** | 55.63 | 49.67 | 45.12 | 83.78 | 67.50 | 63.40 |
| | SPOT (ours) | 5% | 63.53 | 90.82 | 80.69 | 77.91 | 54.82 | 50.22 | 46.38 | 80.80 | 63.53 | 59.31 |
| | SPOT (ours) | 20% | 65.45 | 90.55 | 80.59 | 77.56 | 56.07 | 51.68 | 47.56 | 83.52 | 65.45 | 61.11 |
| | SPOT (ours) | 100% | **67.36** | **90.94** | **81.12** | 78.09 | **57.75** | **53.03** | **47.86** | **87.00** | **67.93** | **63.50** |
| PV-RCNN | From Scratch | - | 66.71 | 91.81 | 82.52 | 80.11 | 58.78 | 53.33 | 47.61 | 86.74 | 64.28 | 59.53 |
| | BEV-MAE (Lin & Wang, 2022) | 100% | 69.91 | 92.55 | 82.81 | 81.68 | 64.82 | 57.13 | 51.98 | 88.22 | 69.78 | 65.75 |
| | AD-PT (Yuan et al., 2023) | 100% | 69.43 | 92.18 | 82.75 | 82.12 | 65.50 | 57.59 | 51.84 | 84.15 | 67.96 | 64.73 |
| | SPOT (ours) | 5% | 70.33 | **92.68** | 83.18 | **82.26** | 63.82 | 56.14 | 51.12 | 89.18 | 71.68 | 67.17 |
| | SPOT (ours) | 20% | 70.85 | 92.61 | 83.06 | 82.03 | 65.66 | 58.02 | 52.55 | **89.77** | 71.48 | **68.01** |
| | SPOT (ours) | 100% | **71.77** | 92.19 | **84.47** | 82.02 | **67.31** | **59.14** | **53.41** | 89.71 | **71.69** | 67.10 |

stream segmentation task, we train 20 and 10 epochs for SemanticKITTI and nuScenes respectively. Our experiments are implemented based on 3DTrans (Team, 2023), using 8 NVIDIA Tesla A100 GPUs. Note that our experiments are under label-efficiency setting, which means that we conduct fine-tuning on a randomly selected subset of the downstream datasets (5% for *NuScenes* detection, 20% for *KITTI* and *ONCE* and 10% for *SemanticKITTI* and *NuScenes* segmentation).

## 4.2 MAIN RESULTS

**NuScenes Detection.** Equipped with different types of LiDAR sensors, the domain gap between the pre-training dataset Waymo and the downstream dataset NuScenes is non-negligible. By harnessing the capabilities of SPOT, which learns general 3D scene representations, it can be found in Tab. 1 that SPOT achieves considerable improvements on the SECOND and CenterPoint detectors compared to other pre-training strategies. Specifically, when pre-trained by 100% Waymo data, SPOT achieves the best overall performance (mAP and NDS) among all the pre-training methods including randomly initialization, BEV-MAE and AD-PT, improving training-from-scratch by up to 10.41 mAPs and 12.69 NDS. Scalable pre-training can also be observed when increasing the amount of pre-training data. When further looking into the detailed categories, SPOT almost achieves the best performance among all the categories for both detectors. For example, SPOT improves SECOND on Bus, Trail, Barries, Motorcycle and Pedestrian for more than 10 mAP compared to training from scratch, which is essential for downstream safety control in real-world deployment.

**KITTI Detection.** Although KITTI uses the same type of LiDAR sensor as that in Waymo dataset, KITTI only employs front-view point clouds for detection, which still introduces domain gaps. In Tab. 2, it can be found that, SECOND and PV-RCNN detectors with SPOT method are significantly and continuously improved as more pre-training data are added. For 100% pre-training data, the improvements are respectively 5.66 and 5.06 mAPs at moderate level. For detailed categories, SPOT brings consistent improvement over different classes. When we focus on Moderate level, the most commonly used metrics, SPOT achieves the best among all the initialization methods for all classes, which shows great potential to avoid disaster in real-world applications.

Table 3: Fine-tuning performance on SemanticKITTI for **segmentation task** using 100% pre-training data. We fine-tune on 10% training data and show the results of some of the categories.

| Backbone | Method | mIOU | car | truck | bus | person | bicyclist | road | fence | trunk |
|---|---|---|---|---|---|---|---|---|---|---|
| Cylinder3D | From Scratch | 49.01 | 93.73 | 38.03 | 25.42 | 35.52 | 0.00 | 92.55 | 46.46 | 65.22 |
| | BEV-MAE (Lin & Wang, 2022) | 53.81 | 94.06 | 58.46 | 38.13 | 50.08 | 51.46 | 92.46 | 46.96 | 62.28 |
| | AD-PT (Yuan et al., 2023) | 52.85 | 94.02 | 42.03 | 36.90 | 50.26 | 49.49 | 91.94 | 49.90 | 60.10 |
| | SPOT (ours) | **55.58** | **94.34** | **61.27** | **43.01** | **55.56** | **67.61** | **92.61** | **52.81** | **67.17** |

Table 4: Fine-tuning performance on NuScenes for **segmentation task** using 100% pre-training data. We fine-tune on 5% and 10% NuScenes training data, respectively, and show the results of some of the categories.

| Backbone | Method | Fine-tuning | mIOU | bus | car | ped. | trailer | sidewalk | vegetable |
|---|---|---|---|---|---|---|---|---|---|
| Cylinder3D | From Scratch | 5% | 45.85 | 10.88 | 75.29 | 47.68 | 15.61 | 61.07 | 80.81 |
| | BEV-MAE (Lin & Wang, 2022) | 5% | 46.94 | 43.48 | 69.68 | 51.63 | 14.04 | 61.27 | 80.42 |
| | AD-PT (Yuan et al., 2023) | 5% | 45.61 | 9.33 | 76.08 | 51.27 | 15.95 | 60.49 | 79.67 |
| | SPOT (ours) | 5% | **49.88** | **50.35** | **76.26** | **52.42** | **16.45** | **63.74** | **81.83** |
| | From Scratch | 10% | 53.72 | 60.54 | 75.28 | 55.90 | 33.47 | 64.02 | 81.62 |
| | BEV-MAE (Lin & Wang, 2022) | 10% | 53.75 | 57.11 | 76.26 | 54.88 | 20.92 | 65.00 | 81.81 |
| | AD-PT (Yuan et al., 2023) | 10% | 52.86 | 53.76 | 81.09 | 53.11 | 28.60 | 65.45 | 82.14 |
| | SPOT (ours) | 10% | **56.10** | **63.24** | **81.30** | **57.86** | **33.99** | **67.04** | **82.73** |

Table 5: The impact of pre-training task superiority. We perform fine-tuning experiments on multiple datasets of both detection and segmentation tasks, using 100% pre-training data.

| Different Pre-training Tasks | KITTI (det) | nuScenes (det) | | SemanticKITTI (seg) | nuScenes (seg) |
|---|---|---|---|---|---|
| | mAP (mod.) | mAP | NDS | mIoU | mIoU |
| Without Pre-training | 61.70 | 42.37 | 52.01 | 60.60 | 69.15 |
| Detection Pre-training | 65.46 | 40.89 | 49.75 | 60.20 | 69.31 |
| Segmentation Pre-training | 58.13 | 36.23 | 47.01 | 61.95 | 69.60 |
| Occupancy Prediction | **67.36** | **47.47** | **57.11** | **62.24** | **70.77** |

**ONCE Detection.** As shown in Fig. 5, when pre-trained by SPOT (solid lines), both SECOND and CenterPoint outperform training from scratch (dot lines) by considerable margins (2.70 and 7.58 mAP respectively). Meanwhile, increasing pre-training data also enlarges this gap, which again demonstrates the ability of SPOT to scale up.

**SemanticKITTI Segmentation.** Results are presented in Tab. 3. It can be found that SPOT significantly improves mIoU metrics compared to training from scratch and achieves the best performance among all pre-training methods. For detailed categories, SPOT gains more than 20 mIoU improvement compared to random initialization on truck, person and bicyclist, which can help guarantee safety in control task.

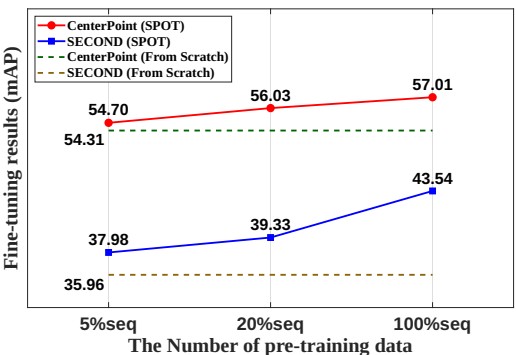

Figure 5: Fine-tuning on ONCE dataset for detection task, where 20% training data are used.

**NuScenes Segmentation.** As shown in Tab. 4, considerable gains are achieved by SPOT, 4.03 and 2.38 mIOUs on 5% and 10% NuScenes data respectively. SPOT also achieves the best performance among all initialization methods.

### 4.3 DISCUSSIONS AND ANALYSES

**Pre-training Tasks.** We argue that occupancy prediction is a scalable and general task for 3D representation learning. Here we conduct experiments to compare different kinds of existing task for pre-training, including detection and segmentation tasks. Pre-training is conducted on the full Waymo dataset and downstream datasets include 20% KITTI data, 5% nuScenes(det) data, 100% SemanticKITTI data, and 100% nuScenes(seg) data. The results presented in Tab. 5 reveal that relying solely on detection as a pre-training task yields minimal performance gains, particularly

Table 6: Ablation study on pre-training strategies across different datasets.

| Occupancy Prediction | Loss Balancing | Beam Re-sampling | Dataset Balancing | nuScenes | | ONCE | KITTI |
|---|---|---|---|---|---|---|---|
| | | | | mAP | NDS | mAP | mAP (mod.) |
| | | | | 32.16 | 41.59 | 35.96 | 61.70 |
| ✓ | | | | 36.55 | 46.98 | 36.00 | 63.70 |
| ✓ | ✓ | | | 37.90 | 47.82 | 37.30 | 64.70 |
| ✓ | ✓ | ✓ | | 38.63 | 48.85 | 39.19 | 65.92 |
| ✓ | ✓ | ✓ | ✓ | **40.39** | **51.65** | **40.63** | **66.45** |

Table 7: Fine-tuning performance on KITTI and NuScenes (det) benchmark with 100% data. We use SECOND as our baseline.

| Method | KITTI | NuScenes (det) | |
|---|---|---|---|
| | mAP(mod.) | mAP | NDS |
| From Scratch | 66.70 | 50.59 | 62.29 |
| SPOT (ours) | **68.57** | **51.88** | **62.68** |

Table 8: Fine-tuning performance on SemanticKITTI and NuScenes (seg) benchmark with 100% data.

| Method | SemanticKITTI | NuScenes (seg) |
|---|---|---|
| | mIOU | mIOU |
| From Scratch | 60.60 | 69.15 |
| SPOT (ours) | **62.24** | **70.77** |

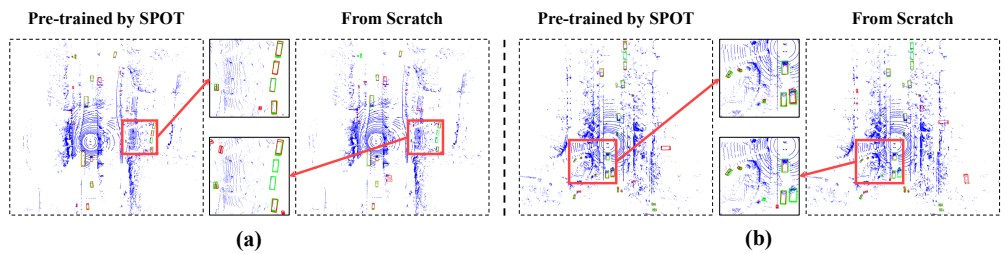

Figure 6: Visualization of downstream results. Red and green boxes are predicted results and the ground truth, respectively.

when significant domain discrepancies exist, *e.g.* Waymo to NuScenes. Similarly, segmentation alone as a pre-training task demonstrates poor performance in the downstream detection task, likely due to the absence of localization information. On the contrary, our occupancy prediction task is beneficial to achieve consistent performance improvements for various datasets and tasks.

**Module-level Ablation Studies in SPOT.** We conduct ablation experiments to analyze the individual components of the proposed SPOT. For pre-training, we uniformly sample 5% Waymo data and subsequently perform fine-tuning experiments on subsets of 5% NuScenes (det) data, 20% KITTI data, and 20% ONCE dataset, using SECOND as the detector. The results presented in Tab. 6 demonstrate the effectiveness of the occupancy prediction task in enhancing the performance of the downstream tasks. Moreover, our proposed strategies for pre-training, including loss balancing, beam re-sampling, and dataset balancing, yield significant improvements in different datasets.

**Beyond the Label-efficiency Setting.** We further conduct experiments on complete downstream datasets, and the results are shown in Tab. 7 and Tab. 8. It can be found that SPOT achieves consistent performance gains even with 100% labeled data, which highlights the effectiveness of SPOT.

## 4.4 QUALITATIVE RESULTS

We fine-tune the model on 20% ONCE training data with SPOT and random initialization, respectively. Fig. 6 showcases the detection results on the validation set, where the red and green boxes correspond to the predicted results and the ground truth, respectively. As shown in the zoom-in areas, it becomes evident that SPOT enhances the ability of SECOND to detect objects located at greater distances, despite these objects having a minimal number of points. More visualization results are illustrated in Figs. 8 and 9 of Appendix C.

## 5 CONCLUSION

In this paper, we introduce SPOT, a scalable and general 3D representation learning method for LiDAR point clouds. SPOT utilizes occupancy prediction as the pre-training task and narrows domain gaps between different datasets by beam re-sampling augmentation and class-balancing strategies. Consistent improvement in various downstream datasets and tasks as well as scalable pre-training are observed. We believe SPOT paves the way for large-scale pre-training on LiDAR point clouds.

**Ethics Statement.** The proposed SPOT focuses on learning general representations via occupancy prediction task, achieving both task- and dataset-level generalization. Besides, SPOT achieves scalable performance gains across different datasets and tasks in label-efficient setting, which is practical in reducing the cost of data acquisition. However, to ensure the safety of self-driving vehicle, a high performance model under the label-efficient setting on multi-tasks needs to be further discussed and studied in the future.

**Reproducibility Statement.** Our work can be verified to be reproducible from the following aspects. (1) In Sec. 4.1, we have clarified the pre-training and downstream datasets employed in SPOT and the corresponding evaluation metrics for each benchmark. (2) We have introduced the baseline model and elaborated on the implementation details, covering all hyper-parameters for both pre-training and fine-tuning stages in Sec. 4.1. (3) We have included the corresponding code files into the supplementary material to further demonstrate the reproducibility of our work.

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

# A    DATASETS DETAILS

**Waymo Open Dataset.** Waymo Open Dataset is a widely used outdoor self-driving dataset, which is collected in multiple cities, namely San Francisco, Phoenix, and Mountain View, using a combination of one 64-beam mid-range LiDAR and four 200-beam short-range LiDARs. This dataset contains a total of 1150 scene sequences, which are further divided into 798 training, 202 validation, and 150 testing sequences. Each sequence spans approximately 20 seconds and consists of around 200 frames of point cloud data, with each point cloud scene covering an area of approximately $150m \times 150m$.

**NuScenes Dataset.** NuScenes Dataset is a highly utilized publicly available dataset in the field of autonomous driving. It encompasses 1000 driving scenarios collected in both Boston and Singapore, with 700 for training, 150 for validation, and 150 sequences for testing. The point cloud data is collected by a 32-beam LiDAR sensor and contains diverse annotations for various tasks, (*e.g.* 3D object detection and 3D semantic segmentation).

**KITTI Dataset.** KITTI dataset, collected in Germany, comprises data captured by a 64-beam LiDAR. It consists of 7481 training samples and 7581 test samples, with the training set further divided into 3712 and 3769 samples for training and validation, respectively. It is worth noting that unlike other datasets, KITTI dataset only provides labels within the front camera field of view.

**ONCE Dataset.** ONCE dataset is a large-scale autonomous dataset collected in China using a 40-beam LiDAR. It encompasses a diverse range of data collected at various times, under different weather conditions, and across multiple regions. The dataset comprises over one million frames of point cloud data, with approximately 15K frames containing annotations. The remaining unlabelled point cloud data serves as resources for unsupervised and semi-supervised algorithms.

**SemanticKITTI Dataset.** SemanticKITTI dataset is a large-scale dataset based on the KITTI Vision Benchmark, collected by a 64-beam LiDAR sensor. It has 22 sequences, of which sequences 0-7 and 9-10 are used as the training set (19K frames in total), and sequence 8 (4K frames) is used as the validation set, and the remaining 11 sequences (20K frames) as the test set.

# B    ADDITIONAL EXPERIMENTS

## B.1    FINE-TUNING PERFORMANCE ON WAYMO DETECTION

We also perform detailed experiments in the downstream Waymo detection task. We evaluate the results using the official Average Precision (AP) and Average Precision with Heading (APH), with a particular focus on the more challenging L2-LEVEL metrics. The evaluation results on the Waymo validation set are presented in Tab. 9. We conduct fine-tuning on 3% data using the widely adopted CenterPoint detector. Furthermore, we confirm the scalability of SPOT and achieve superior performance compared to training from scratch. Specifically, SPOT improves the performance of training from scratch by 4.76 and 4.69 for CenterPoint in L2 AP and L2 APH. Tab. 9 illustrates that SPOT with only 5% sequence-level pre-training data can outperform BEV-MAE and AD-PT using 100% pre-training data.

Table 9: Fine-tuning performance on Waymo benchmark (LEVEL_2 metric). We fine-tune on 3% Waymo training data. P.D.A. represents the Pre-training Data Amount.

| Backbone | Method | P.D.A. | L2 AP / APH | | | |
|---|---|---|---|---|---|---|
| | | | Overall | Vehicle | Pedestrian | Cyclist |
| CenterPoint | From Scratch | - | 59.00 / 56.29 | 57.12 / 56.57 | 58.66 / 52.44 | 61.24 / 59.89 |
| | BEV-MAE (Lin & Wang, 2022) | 100% | 59.51 / 56.81 | 57.38 / 56.84 | 58.87 / 52.78 | 62.28 / 60.82 |
| | AD-PT (Yuan et al., 2023) | 100% | 61.21 / 58.46 | 60.35 / 59.79 | 60.57 / 54.02 | 62.73 / 61.57 |
| | SPOT (ours) | 5% | 61.61 / 58.69 | 58.63 / 58.06 | 61.35 / 54.53 | 64.86 / 63.48 |
| | SPOT (ours) | 20% | 62.74 / 59.84 | 59.67 / 59.09 | 62.73 / 56.01 | 65.83 / 64.41 |
| | SPOT (ours) | 100% | **63.76 / 60.98** | **61.17 / 60.63** | **64.05 / 57.49** | **66.07 / 64.81** |

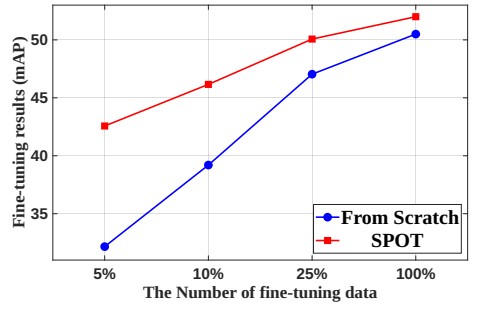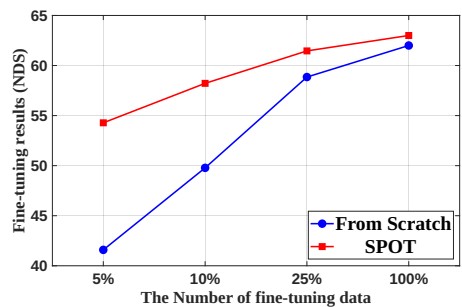

(a) Fine-tuning results on nuScenes using mAP metric.

(b) Fine-tuning results on nuScenes using NDS metric.

Figure 7: Fine-tuning performance on nuScenes dataset for detection task with different numbers of annotated data.

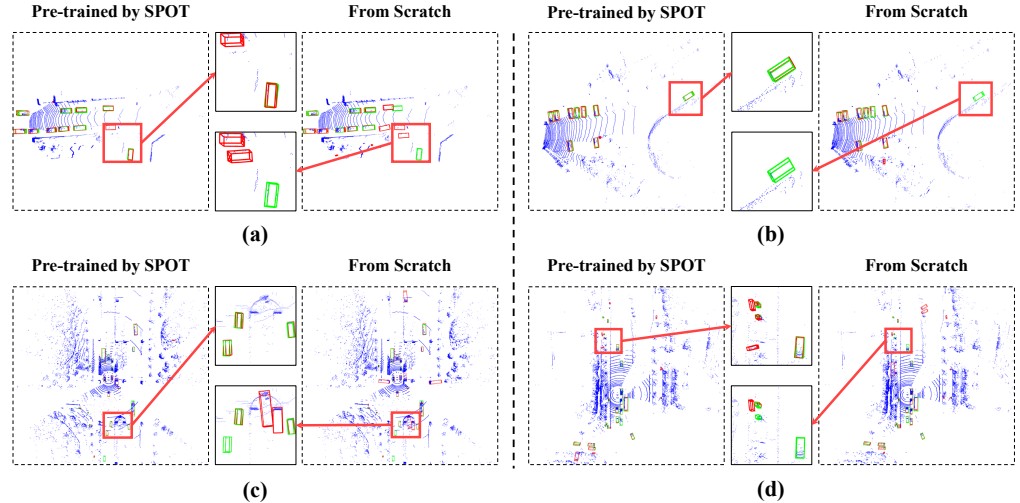

Figure 8: Visualization of downstream detection results, where the red and green boxes correspond to the predicted results and the ground truth, respectively. (a) and (b) are the results of KITTI, (c) and (d) are the results of ONCE.

### B.2 DATA-EFFICIENCY FOR DOWNSTREAM

In order to illustrate the influence of the pre-training method on downstream data, we conduct the fine-tuning experiments on nuScenes dataset using varying proportions of annotated data (*e.g.*, 5%, 10%, 25%, and 100% budgets), using SECOND as the detector. Fig. 7 shows the results of our experiments, highlighting the consistent performance improvement achieved by SPOT across different budget allocations, demonstrating its effectiveness in improving data efficiency.

### B.3 PRE-TRAINING ON NUSCENES

To verify that SPOT is able to pre-train on other datasets, we utilize the model which is pre-trained on Waymo to predict occupancy labels on 5% NuScenes dataset and generate pseudo occupancy labels. Next, we pre-train SPOT from scratch on such NuScenes data, and then fine-tune on the 20% KITTI data. As shown in Tab. 10, SPOT achieves significant gains compared to baseline results on KITTI dataset, demonstrating the effectiveness and generalization of SPOT.

### B.4 SEMI-SUPERVISED PRE-TRAINING SETTING

Table 10: Pre-training on NuScenes dataset and fine-tuning on KITTI benchmark for detection task. We fine-tune on 20% training data.

| Backbone | Method | F.D.A. | mAP |
|---|---|---|---|
| SECOND | From Scratch | 20% | 61.70 |
| | SPOT (ours) | 20% | **64.39** |
| PV-RCNN | From Scratch | 20% | 66.71 |
| | SPOT (ours) | 20% | **69.58** |

Table 11: Semi-supervised pre-training setting. We fine-tune using 5% training data on NuScenes benchmark for detection task. L5% denotes that we perform the pre-training on 5% sequence-level labeled data, while U5% represents the 5% unlabeled data.

| Backbone | Method | P.D.A. | F.D.A. | mAP | NDS |
|---|---|---|---|---|---|
| SECOND | From Scratch | - | 5% | 32.16 | 41.59 |
| | SPOT | L5% | 5% | 37.96 | 48.45 |
| | SPOT | L20% | 5% | 39.63 | **51.63** |
| | SPOT (SEMI) | L5% + U5% | 5% | 38.50 | 50.03 |
| | SPOT (SEMI) | L5% + U15% | 5% | **39.81** | 51.51 |
| CenterPoint | From Scratch | - | 5% | 42.37 | 52.01 |
| | SPOT | L5% | 5% | 43.56 | 53.04 |
| | SPOT | L20% | 5% | 44.94 | 54.95 |
| | SPOT (SEMI) | L5% + U5% | 5% | 43.65 | 53.82 |
| | SPOT (SEMI) | L5% + U15% | 5% | **45.18** | **54.98** |

Table 12: Semi-supervised pre-training setting. We fine-tune using 20% training data on KITTI benchmark for detection task.

| Backbone | Method | P.D.A. | F.D.A. | mAP |
|---|---|---|---|---|
| SECOND | From Scratch | - | 20% | 61.70 |
| | SPOT | L5% | 20% | 63.53 |
| | SPOT | L20% | 20% | 65.45 |
| | SPOT (SEMI) | L5% + U5% | 20% | 65.18 |
| | SPOT (SEMI) | L5% + U15% | 20% | **66.45** |
| PV-RCNN | From Scratch | - | 20% | 66.71 |
| | SPOT | L5% | 20% | 70.33 |
| | SPOT | L20% | 20% | 70.85 |
| | SPOT (SEMI) | L5% + U5% | 20% | 70.40 |
| | SPOT (SEMI) | L5% + U15% | 20% | **70.86** |

Table 13: Semi-supervised pre-training setting. We fine-tune using 20% training data on ONCE benchmark for detection task.

| Backbone | Method | P.D.A. | F.D.A. | mAP |
|---|---|---|---|---|
| SECOND | From Scratch | - | 20% | 35.96 |
| | SPOT | L5% | 20% | 37.98 |
| | SPOT | L20% | 20% | 39.33 |
| | SPOT (SEMI) | L5% + U5% | 20% | 38.31 |
| | SPOT (SEMI) | L5% + U15% | 20% | **39.85** |
| CenterPoint | From Scratch | - | 5% | 54.31 |
| | SPOT | L5% | 5% | 54.70 |
| | SPOT | L20% | 5% | **56.03** |
| | SPOT (SEMI) | L5% + U5% | 20% | 55.05 |
| | SPOT (SEMI) | L5% + U15% | 20% | 56.00 |

Although it has been observed that SPOT learns scalable and general 3D representation for various datasets and tasks, labeling burden from occupancy prediction still brings concerns on its scalability. Thus, we perform experiments to explore the semi-supervised setting during pre-training phase to further demonstrate the ability of SPOT to scale up. Specifically, we pre-train the 3D/2D backbones with SPOT using only 5% sequence-level labeled data and 15% sequence-level unlabeled data, where the unlabeled data is pseudo-labeled (Lee et al., 2013) by employing a naive mean-teacher approach (Tarvainen & Valpola, 2017). The pre-trained 3D/2D backbones are fine-tuned on different downstream tasks including NuScenes, KITTI, ONCE detection tasks and NuScenes and SemanticKITTI segmentation tasks using different baseline models, as shown in Tab. 11, Tab. 12, Tab. 13, Tab. 14, Tab. 15. It can be found that semi-supervised pre-training with SPOT achieves comparable downstream performance as that of fully-supervised pre-training. It consistently improve different architectures on various datasets and tasks. Also, when incorporating more unlabeled data, the performance improves. Thus, we believe that SPOT is able to generalize to label-efficient pre-training setting and further attain the performance scalability on different downstream datasets and tasks such as 3D detection and segmentation tasks.

## B.5 GENERALIZABILITY FOR TRANSFORMER-BASED STRUCTURE

In order to further verify the generalizability of our approach towards the Transformer-based network structure, we have conducted experiments on DSVT model Wang et al. (2023). First, we employ the encoder of DSVT model Wang et al. (2023) and perform the pre-training process using SPOT on 20% sequence-level data from Waymo. Then, the fine-tuning experiments are conducted on the NuScenes and ONCE datasets. The results shown in Tab. 16 demonstrate that, for the transformer-based baseline, SPOT also achieves significant gains under different benchmarks.

## B.6 EXPERIMENTS ON EXTENDING THE TRAINING SCHEDULE

To further demonstrate that our pre-training method strengthen the backbone capacity rather than simply accelerating the convergence speed of training model, we consider conducting experiments under different training schedules. We select SECOND, CenterPoint, and DSVT, as the baseline method, and the experimental results are shown in Tab. 17. It can be seen from these results that, the

Table 14: Semi-supervised pre-training setting. We fine-tune using 5% training data on NuScenes benchmark for segmentation task.

| Backbone | Method | P.D.A. | F.D.A. | mIOU |
|---|---|---|---|---|
| | From Scratch | - | 5% | 45.85 |
| | SPOT | L5% | 5% | 46.71 |
| Cylinder3D | SPOT | L20% | 5% | 47.84 |
| | SPOT (SEMI) | L5% + U5% | 5% | 47.60 |
| | SPOT (SEMI) | L5% + U15% | 5% | **48.84** |

Table 15: Semi-supervised pre-training setting. We fine-tune using 10% training data on SemanticKITTI benchmark for segmentation task.

| Backbone | Method | P.D.A. | F.D.A. | mIOU |
|---|---|---|---|---|
| | From Scratch | - | 10% | 49.01 |
| | SPOT | L5% | 10% | 52.50 |
| Cylinder3D | SPOT | L20% | 10% | 54.10 |
| | SPOT (SEMI) | L5% + U5% | 10% | 53.62 |
| | SPOT (SEMI) | L5% + U15% | 10% | **54.70** |

Table 16: Fine-tuning performance employing transformer-based structure on different datasets.

| Detector | Method | P.D.A. | NuScenes | | ONCE |
|---|---|---|---|---|---|
| | | | mAP | NDS | mAP |
| | From Scratch | - | 49.78 | 58.63 | 51.52 |
| DSVT | SPOT (ours) | 5% | 55.47 | 62.17 | 57.81 |
| | SPOT (ours) | 20% | **56.65** | **63.52** | **59.78** |

results of only training 30 epochs using our SPOT pre-training can exceed the results of 150 epochs of training from scratch by $2.35\% \sim 5.78\%$.

### B.7 EXPERIMENTS USING DIFFERENT SAMPLING STRATEGIES ON DOWNSTREAM TASK.

We also try some other subset sampling methods in downstream tasks, such as the uniform sampling method, and give experimental results in NuScenes segmentation task. Tab. 18 shows that, SPOT can still achieve consistent performance improvement under the uniform sampling strategy. In the future, we will try more sampling strategies to help downstream tasks get better performance.

### B.8 PRE-TRAINING WITH BINARY OCCUPANCY LABELS

We conduct additional experiments using 20% sequence-level binary occupancy-based Waymo data to perform the pre-training, and employ 5% NuScenes data for downstream fine-tuning. For consistency with previous experiments, we use the widely-adopted CenterPoint as our baseline model. The experimental results are shown in the following Tab. 19. It can be seen that, simple binary occupancy prediction does not bring performance gains when it performs cross-domain experiments, such as Waymo to NuScenes. This is mainly due to that, the pre-training model is difficult to learn semantically-rich information of the 3D scene only employing binary occupancy prediction as pre-training task.

## C  VISUALIZATION RESULTS

Firstly, Fig. 8 shows the visualization results of different downstream datasets (*i.e.*, KITTI, ONCE). The visualization results of different downstream datasets also demonstrate that our SPOT boosts the ability of the baseline for 3D object detection task compared to training from scratch.

Secondly, Fig. 9 presents the visualization of the results obtained from our pre-training task on the Waymo validation set, showcasing the raw input point cloud on the left, while the middle and right sections display our predicted occupancy results and the Ground Truth (GT) of the dataset, respectively. Fig. 9 clearly demonstrates our ability to generate highly dense occupancy prediction using a sparse single-frame point cloud input. Furthermore, it is worth noting that the occupancy GT also exhibits sparsity in certain areas, such as certain sections of the road surface. This sparsity is inherent to LiDAR sensor, as there will always be some areas that are not scanned and virtually have no points in the frame. However, our prediction results exhibit greater continuity and produce superior performance in these details, which confirms the scene understanding capability of SPOT.

Table 17: Experiments of extending the training schedule on NuScenes for detection task.

| Detector | Method | P.D.A. | Training Schedule | mAP | NDS |
|---|---|---|---|---|---|
| SECOND | From Scratch | - | 30 epochs | 32.16 | 41.59 |
| | From Scratch | - | 150 epochs | 36.79 | 51.01 |
| | SPOT (ours) | 20% | 30 epochs | 39.63 | 51.63 |
| | SPOT (ours) | 100% | 30 epochs | **42.57** | **54.28** |
| CenterPoint | From Scratch | - | 30 epochs | 42.37 | 52.01 |
| | From Scratch | - | 150 epochs | 41.01 | 53.92 |
| | SPOT (ours) | 20% | 30 epochs | 44.94 | 54.95 |
| | SPOT (ours) | 100% | 30 epochs | **47.47** | **57.11** |
| DSVT | From Scratch | - | 20 epochs | 49.78 | 58.63 |
| | From Scratch | - | 150 epochs | 54.30 | **63.58** |
| | SPOT (ours) | 20% | 20 epochs | **56.65** | 63.52 |

Table 18: Fine-tuning on NuScenes benchmark for segmentation task using different sampling strategies. We fine-tune on 10% training data.

| Method | Sampling Strategy | F.D.A. | mIOU |
|---|---|---|---|
| From Scratch (Cylinder3D) | random | 10% | 53.72 |
| SPOT (Cylinder3D) | random | 10% | **56.10** |
| From Scratch (Cylinder3D) | uniform | 10% | 52.96 |
| SPOT (Cylinder3D) | uniform | 10% | **56.14** |

Table 19: Fine-tuning performance on NuScenes benchmark for detection task based on the binary occupancy pre-training. We fine-tune on 5% training data.

| Backbone | Method | F.D.A. | mAP | NDS |
|---|---|---|---|---|
| CenterPoint | From Scratch | 5% | 42.37 | 52.01 |
| | Binary occupancy pre-training | 5% | 42.05 | 51.63 |
| | SPOT (ours) | 5% | **44.94** | **54.95** |

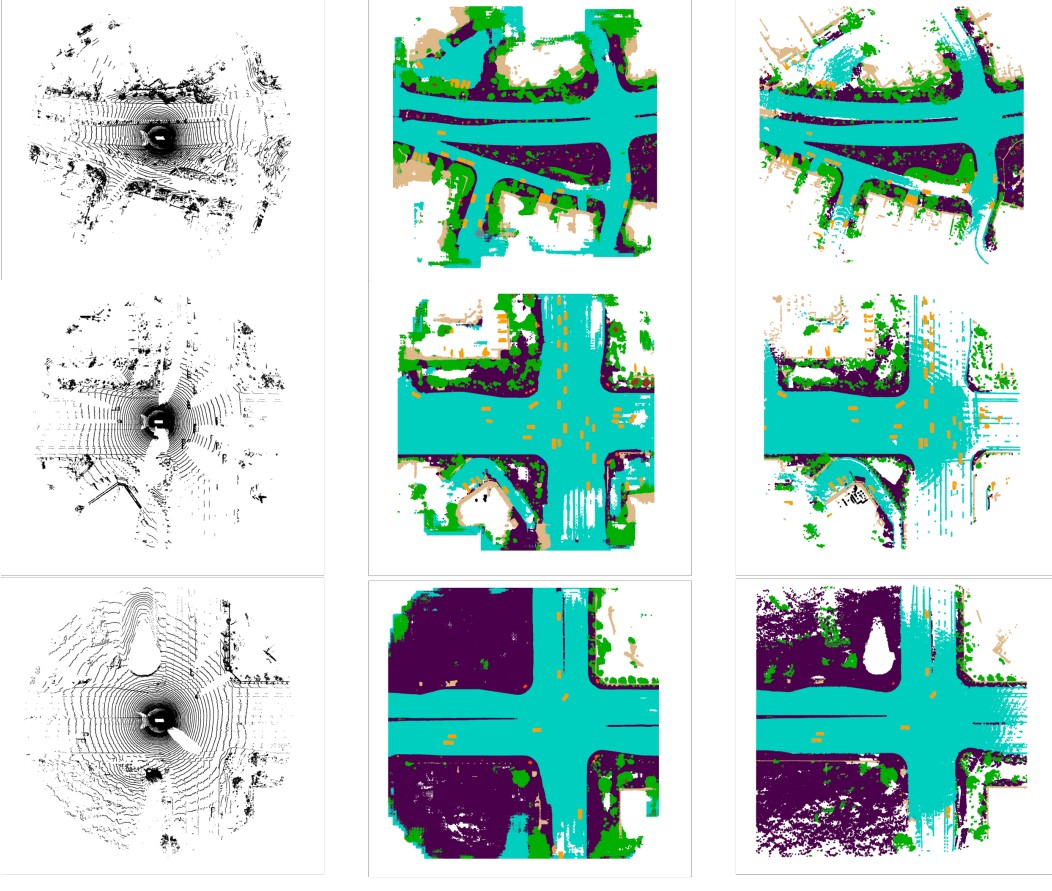

Input Points      Occupancy Prediction      Occupancy GT

Figure 9: Visualization results of occupancy prediction on Waymo validation set.

