# OpenReview forum: "SPOT: Scalable 3D Pre-training via Occupancy Prediction for Autonomous Driving"
_ICLR.cc/2024/Conference — Submitted to ICLR 2024_

### Official Review · Reviewer_FWLb · 2023-10-13

**Soundness:** 3 good
**Presentation:** 4 excellent
**Contribution:** 3 good
**Rating:** 5
**Confidence:** 4

**Summary:**

The paper introduces SPOT (Scalable Pretraining via Occupancy prediction for learning Transferable 3D representations), a method for easing the annotation of 3D LiDAR point clouds, which is typically resource-intensive. The central idea is to leverage large-scale pre-training and then refine these models for different downstream tasks and datasets. Main contributions can be summarised as follows:

- Occupancy Prediction: The paper underscores the potential of occupancy prediction as a means to learn general representations. The efficacy of this approach is validated through comprehensive experiments on numerous datasets and tasks.
- Techniques for Point Cloud Augmentation: SPOT employs a beam re-sampling method to augment point clouds. It also implements class-balancing strategies to counteract the disparities arising from diverse LiDAR sensors and annotation methodologies across datasets.
- Scalability of Pre-training: An interesting observation made is that as the amount of pre-training data increases, the performance on downstream tasks improves consistently.

**Strengths:**

- The paper addresses a significant challenge in 3D LiDAR point cloud research, specifically the task of annotating 3D LiDAR point clouds for perception. The approach of performing large-scale pre-training and then fine-tuning the pre-trained backbone on various downstream datasets is novel.

- Comprehensive experiments on multiple pre-training and downstream datasets (WOD, KITTI, SemanticKITTI, NuScenes, ONCE) are presented, along with thorough ablation studies.

- The structure and writing of the manuscript are clear, making it easy to follow.

- The figures, visualizations and illustrations are exemplary, with a particular appreciation for Fig. 1.

**Weaknesses:**

- Some of the contributions highlighted by the authors appear to be not novel enough, e.g., the class-balancing strategies. Existing studies [1, 2], have showcased similar strategies. It would be good to acknowledge, cite, and compare their work with these prior studies.

- In Sec. 4.1, the authors claim that "our experiments are under label-efficiency setting." However, contemporary dataset subsampling techniques seem to encompass more than just the "randomly selected" method utilised in this paper. In fact, drawing from semi-supervised methodologies, the "uniform sampling" technique appears to be more prevalent. Moreover, Li et al. introduced the ST-RFD method, which aims to extract a more diverse subset of training data frame samples. I believe it would be beneficial for the authors to explore different sampling techniques and consider integrating the ST-RFD method to potentially achieve better downstream subset(s).

- While the authors introduce a two-stage approach of first pre-training followed by fine-tuning, I'm still uncertain about the main advantages of that idea. For instance, current research employing semi-supervised or active learning techniques for point cloud semantic segmentation [3, 4] seems to achieve superior results  on 10% of the SemanticKITTI dataset (mIoU: 62.2 [3], mIoU: 60.0 [4]). Furthermore, Unal et al. [2] obtain a 61.3 mIoU on ScribbleKITTI (weakly annotated with 8% of the SemanticKITTI points). These methods also seem to efficiently utilise labels and alleviate the burden of extensive labelling.

- Lack of related work, e.g., [1-4].

[1] Zou, Y., Yu, Z., Kumar, B. V. K., & Wang, J. (2018). Unsupervised domain adaptation for semantic segmentation via class-balanced self-training. In Proceedings of the European conference on computer vision (ECCV) (pp. 289-305).

[2] Unal, O., Dai, D., & Van Gool, L. (2022). Scribble-supervised lidar semantic segmentation. In *Proceedings of the IEEE/CVF Conference on Computer Vision and Pattern Recognition* (pp. 2697-2707).

[3] Li, L., Shum, H. P., & Breckon, T. P. (2023). Less is more: Reducing task and model complexity for 3d point cloud semantic segmentation. In *Proceedings of the IEEE/CVF Conference on Computer Vision and Pattern Recognition* (pp. 9361-9371).

[4] Kong, L., Ren, J., Pan, L., & Liu, Z. (2023). Lasermix for semi-supervised lidar semantic segmentation. In *Proceedings of the IEEE/CVF Conference on Computer Vision and Pattern Recognition* (pp. 21705-21715).

**Questions:**

Refer to Weaknesses

---

> ### Author Response · Authors · 2023-11-15
> **Response to Reviewer FWLb**
>
> Dear Reviewer FWLb,
>
> Thanks for your precious time and considerate suggestions to improve our manuscript.
>
> **Q1: Discussion about the class-balancing strategies with existing studies [1] [2].**
>
> A1:  Thanks for your comments, both methods present excellent self-training category balancing methods. In the process of self-training, Class-balanced self-training (CBST) [1] sets different thresholds for different classes, and then balances classes according to the confidence of false tags, so that false tags with low confidence of rare classes can participate in the training. Based on the CBST, Class-range-balanced Self-training (CRB-ST) [2] adds range-based considerations,  and sample pseudo-labels both from both category and range dimensions. Unlike them, we are mainly from a global perspective to alleviate the problem of class imbalance in the pre-training process. First, to ensure the integrity of occupancy labels in the scene, class sampling is based on frame-level. Secondly, considering that foreground information is more important than background information, and a large amount of background information can easily drown out scarce foreground information, we only rebalance for foreground categories, which can bring benefits to both downstream detection and segmentation tasks.
>
> **Q2: Discussion about using different sampling strategies in downstream task to potentially achieve better downstream subsets.**
>
> A2:  We strongly agree with you that better sampling for downstream datasets leads to better performance. Since we do not use unlabeled data in the downstream dataset, we only perform fine-tuning experiments and fair comparison experiments on the same batch of randomly sampled data. Also, we have added some experimental results with uniform sampling. In the future, we will also integrate different sampling methods, such as the ST-RFD method [3], into the fine-tuning pipeline for better performance.
>
> Table 19. Different sampling strategies on downstream task.
>
> | Method | Sampling Strategy | F.D.A.  | mIOU |
> | --- | --- | --- | --- |
> | From Scratch (Cylinder3D) | random  | 10% NuScenes | 53.72 |
> | SPOT (Cylinder3D) | random  | 10% NuScenes | 56.10 |
> | From Scratch (Cylinder3D) | uniform  | 10% NuScenes | 52.96 |
> | SPOT (Cylinder3D) | uniform  | 10% NuScenes | 56.14 |
>
> **Q3: Discussion about task-specific semi-supervised learning/active learning [2,3,4].**
>
> A3:  [2, 3, 4] are excellent semi-supervised or weakly supervised methods in the field of 3D point cloud segmentation, and they use the labeled data and other pseudo-labels of unlabeled data to achieve excellent performance, but most of these semi-supervised methods are task-specific or dataset-specific, e.g., they can only work in segmentation tasks or detection tasks. In contrast, SPOT focuses on the pre-training part, aiming to bring better transferable prior knowledge to all other tasks and datasets through just one pre-training process, which is decoupled from the downstream datasets and tasks. Moreover, we have not used downstream unlabeled data, meaning that we can combine SPOT with various semi-supervised or weakly-supervised methods in downstream fine-tuning to further improve the performance, by using SPOT as the downstream initialization. We think it's going to be a very effective combination.
>
> **Q4: Lack of related work, e.g., [1-4].**
>
> A4:  Thank you for pointing these out! We has added all of them in the revision.
>
> We hope our reply and additional experiment results address your concerns. Look forward to further discussion on any remaining concern about our paper.
>
> Best regards,
>
> Authors of Paper 496
>
> [1] Zou, Y., Yu, Z., Kumar, B. V. K., & Wang, J. (2018). Unsupervised domain adaptation for semantic segmentation via class-balanced self-training. In Proceedings of the European conference on computer vision (ECCV) (pp. 289-305).
>
> [2] Unal, O., Dai, D., & Van Gool, L. (2022). Scribble-supervised lidar semantic segmentation. In Proceedings of the IEEE/CVF Conference on Computer Vision and Pattern Recognition (pp. 2697-2707).
>
> [3] Li, L., Shum, H. P., & Breckon, T. P. (2023). Less is more: Reducing task and model complexity for 3d point cloud semantic segmentation. In Proceedings of the IEEE/CVF Conference on Computer Vision and Pattern Recognition (pp. 9361-9371).
>
> [4] Kong, L., Ren, J., Pan, L., & Liu, Z. (2023). Lasermix for semi-supervised lidar semantic segmentation. In Proceedings of the IEEE/CVF Conference on Computer Vision and Pattern Recognition (pp. 21705-21715).

---

> ### Author Response · Authors · 2023-11-22
> **Look forward to further discussion.**
>
> Dear Reviewer FWLb,
>
> Thanks again for your great efforts and constructive advice in reviewing this paper! As the deadline of discussion period is approaching, we sincerely hope that our response can address your concerns and we are looking forward to further discussion on any other questions or issues regarding the paper or our response.
>
> Best regards,
>
> Authors of Paper 496

---

### Official Review · Reviewer_4p2k · 2023-10-29

**Soundness:** 2 fair
**Presentation:** 3 good
**Contribution:** 2 fair
**Rating:** 5
**Confidence:** 4

**Summary:**

This paper proposes a scalable pre-training method called SPOT for learning transferable representations for 3D perception tasks. SPOT pre-trains a model on the task of occupancy prediction, which is a general task that can be used to learn useful representations for various 3D perception tasks. To mitigate the gaps between pre-training and fine-tuning datasets, SPOT uses beam resampling augmentation and class-balancing strategies. The authors evaluate SPOT on various 3D perception tasks and datasets. SPOT outperforms training from scratch by a significant margin on all tasks.

**Strengths:**

This paper is well-written and presents extensive studies on pre-training occupancy prediction on the Waymo Open Dataset and fine-tuning on Nuscenes, KITTI, ONCE, and SemanticKitti. The authors make the interesting observation that occupancy prediction outperforms detection pre-training, even on downstream detection tasks. This highlights the ability of occupancy prediction to mitigate the domain gap between pre-training and fine-tuning datasets.

**Weaknesses:**

1. Unfair comparisons in Table 1 and 2: SPOT is compared against BEV-MAE and AD-PT, which are self-supervised and semi-supervised pre-training methods, respectively. This is an unfair comparison, as SPOT is a supervised pre-training method that benefits from human-labeled data. A more fair comparison would be to compare SPOT against other supervised pre-training methods.

2. Supervised occupancy prediction is not new, and It is hard to argue scalability when pre-training is based on a labeled dataset.  On the other hand, self-supervised occupancy pretrainig was demonstrated in previous works, e.g. [1], which had shown unlabeled occupancy pretraining via MAE work.

3. Another critical limitation on scalability is that the pre-training dataset must have a more expensive lidar setup (64-beam with high density) compared to the fine-tuning dataset.

3. Low train from scratch baseline performance: As shown in Fig 7, with 100% nuScenes, training from scratch leads to mAP ~ 50, which is far from SOTA (>70) [2].

4.  Given the low train from scratch performance, could the baseline model be undertrained or that it is using a weaker data augmentation?

Pre-training 30 epochs on WOD requires a lot more computation resources compared to fine-tuning 30 epochs on NuScenes.

If you use the same computation resources ( as pre-training + fine-tuning) to train a model on NuScenes from scratch with a stronger data augmentation, I guess the performance gap between pre-training and training from scratch will be much smaller.

Also, previous studies have shown that with strong data augmentation during fine-tuning, the benefit from pretrianing diminishes [3].

[1] Min, Chen, et al. "Occupancy-MAE: Self-Supervised Pre-Training Large-Scale LiDAR Point Clouds With Masked Occupancy Autoencoders." IEEE Transactions on Intelligent Vehicles (2023).

[2] https://paperswithcode.com/sota/3d-object-detection-on-nuscenes

[3] Zoph, Barret, et al. "Rethinking pre-training and self-training." Advances in neural information processing systems 33 (2020): 3833-3845.

**Questions:**

Please see weaknesses.



-------------------------------------
Thank you for running the additional experiments. I increased my score.

However, I cannot recommend acceptance
1.  the improvement is marginal (only + 0.25 NDS) for a stronger baseline (VoxelNext), Table 17.
2.  As reviewer y7L9 suggested, `the improvements and results showcased appear anticipated`.

---

> ### Author Response · Authors · 2023-11-15
> **Response to Reviewer 4p2k (Part1)**
>
> Dear Reviewer 4p2k,
>
> Thanks for your time and constructive suggestions! We provide additional experiments on semi-supervised pre-training via SPOT and make more fair comparisons to the baselines. Also a detailed discussion on the results are presented.
>
> **Q1: Unfair comparisons with BEV-MAE and AD-PT**
>
> **A1:** The motivation of SPOT is to pre-train a general backbone for different datasets and tasks, which is not achieved by BEV-MAE and AD-PT. For BEV-MAE, they are only able to pre-train and fine-tune on the same dataset. Tables 11, 12, 13 and 14 below show this phenomenon. For AD-PT, fine-tuning on semantic segmentation task harm the performance. Tables 12 and 13 show that when fine-tuning on LiDAR semantic segmentation task, AD-PT brings smaller improvement or even harm the performance.
>
> To make more fair comparisons to AD-PT and also demonstrate the scalability of SPOT, we perform semi-supervised pre-training with SPOT. Results are provided in Tables 11, 12, 13 and 14 below. It can be found that pre-training with 5% sequence-level labeled data and 15% sequence-level unlabeled data achieves comparable downstream fine-tuning performance as that of pre-training with 20% sequence-level labeled data. Although 20% data in Waymo is much less than that used in AD-PT, it can be found that SPOT still achieve comparable or surpass AD-PT in various downstream datasets and tasks. We are also running experiments with more unlabeled data but due to the time limitation the experiments are not finished yet. We would update the results as soon as they finish. However, scalability of SPOT can also be discovered when we compare pre-trainings with different amount of unlabeled data (5% and 15%). It can be found that more unlabeled data brings better downstream performance. These results verify that (1) SPOT can learn general representations for different downstream datasets, tasks and architectures with small part of labels. (2) When incorporating more unlabeled data into pre-training phase, SPOT also shows scalability.
>
> Table 11. Comparison to different pre-training methods. The downstream task is 3D object detection on NuScenes. We provide additional semi-supervised pre-training via SPOT here.
>
> | Backbone    | Method | Pre-training Setting | Pre-training Data Amount | F.D.A.  | mAP  | NDS |
> | --- | --- | --- | --- | --- | --- | --- |
> | SECOND | From Scratch | - | - | 5% Nuscenes | 32.16 | 41.59 |
> | SECOND | BEV-MAE | Unsupervised | 100% | 5% Nuscenes | 32.09 | 42.88 |
> | SECOND | AD-PT | Semi-supervised | 100% | 5% Nuscenes | 37.69  | 47.95 |
> | SECOND | SPOT (SEMI) | Semi-supervised | 20% | 5% Nuscenes | 39.81 | 51.51 |
> | SECOND | SPOT | Fully-supervised | 20% | 5% Nuscenes | 39.63 | 51.63 |
> | CenterPoint | From Scratch | - | - | 5% Nuscenes | 42.37  | 52.01 |
> | CenterPoint | BEV-MAE | Unsupervised | 100% | 5% Nuscenes | 42.86  | 52.95 |
> | CenterPoint | AD-PT | Semi-supervised | 20% | 5% Nuscenes | 44.99  | 52.99 |
> | CenterPoint | SPOT (SEMI) | Semi-supervised | 20% | 5% Nuscenes | 45.18 | 54.98 |
> | CenterPoint | SPOT  | Fully-supervised | 20% | 5% Nuscenes | 44.94 | 54.95 |
>
> Table 12. Comparison to different pre-training methods. The downstream task is 3D object detection on KITTI. We provide additional semi-supervised pre-training via SPOT here.
>
> | Backbone    | Method | Pre-training Setting | Pre-training Data Amount | F.D.A.  | mAP  |
> | --- | --- | --- | --- | --- | --- |
> | SECOND | From Scratch | - | - | 20% KITTI | 61.70 |
> | SECOND | BEV-MAE | Unsupervised | 100% | 20% KITTI | 63.45 |
> | SECOND | AD-PT | Semi-supervised | 100% | 20% KITTI | 65.95 |
> | SECOND | SPOT (SEMI) | Semi-supervised | 20% | 20% KITTI | 65.45 |
> | SECOND | SPOT | Fully-supervised | 20% | 20% KITTI | 66.45 |
> | PV-RCNN | From Scratch | - | - | 20% KITTI | 66.71 |
> | PV-RCNN | BEV-MAE | Unsupervised | 100% | 20% KITTI | 69.91 |
> | PV-RCNN | AD-PT | Semi-supervised | 20% | 20% KITTI | 69.43 |
> | PV-RCNN | SPOT (SEMI) | Semi-supervised | 20% | 20% KITTI | 70.86 |
> | PV-RCNN | SPOT  | Fully-supervised | 20% | 20% KITTI | 70.85 |
>
> Table 13. Comparison to different pre-training methods. The downstream task is LiDAR semantic segmentation on NuScenes. We provide additional semi-supervised pre-training via SPOT here.
>
> | Backbone    | Method | Pre-training Setting | Pre-training Data Amount | F.D.A.  | mIOU |
> | --- | --- | --- | --- | --- | --- |
> | Cylinder3D | From Scratch | - | - | 5% Nuscenes | 45.85 |
> | Cylinder3D | BEV-MAE | Unsupervised | 100% | 5% Nuscenes | 46.94 |
> | Cylinder3D | AD-PT | Semi-supervised | 100% | 5% Nuscenes | 45.61 |
> | Cylinder3D | SPOT (SEMI) | Semi-supervised | 20% | 5% Nuscenes | 48.84 |
> | Cylinder3D | SPOT | Fully-supervised | 20% | 5% Nuscenes | 47.84 |

---

> > ### Author Response · Authors · 2023-11-15
> > **Response to Reviewer 4p2k (Part2)**
> >
> > Table 14. Comparison to different pre-training methods. The downstream task is LiDAR semantic segmentation on SemanticKITTI. We provide additional semi-supervised pre-training via SPOT here.
> >
> > | Backbone    | Method | Pre-training Setting | Pre-training Data Amount | F.D.A.  | mIOU |
> > | --- | --- | --- | --- | --- | --- |
> > | Cylinder3D | From Scratch | - | - | 10% SemanticKITTI   | 49.01 |
> > | Cylinder3D | BEV-MAE | Unsupervised | 100% | 10% SemanticKITTI   | 53.81 |
> > | Cylinder3D | AD-PT | Semi-supervised | 100% | 10% SemanticKITTI   | 52.85 |
> > | Cylinder3D | SPOT (SEMI) | Semi-supervised | 20% | 10% SemanticKITTI   | 54.70 |
> > | Cylinder3D | SPOT | Fully-supervised | 20% | 10% SemanticKITTI   | 54.10 |
> >
> > Thus, semi-supervised occupancy prediction is promising in LiDAR perception pre-training and we believe that SPOT would pave the way for future exploration in LiDAR pre-training.
> >
> > **Q2: A more fair comparison would be to compare SPOT against other supervised pre-training methods**
> >
> > **A2:**  For a fair comparison with other supervised pre-training methods, we pre-train the backbone using object detection or semantic segmentation as pre-training task, and fine-tune on different downstream datasets and tasks. Note that we **use 100% labeled data for pre-training when use detection and segmentation as pre-training tasks**., but **only use 5% sequence-level labeled data and 15% sequence-level unlabeled data for SPOT.**  As shown in Table 15, by comparing 3D detection (or segmentation) pre-training and our proposed occupancy generation prediction, our method yields 2.8% performance gain. This is because in the 3D perception task, there are huge domain differences between different datasets and tasks, and it is difficult to bring benefits from direct adaptation, which also validates the superiority of our approach.
> >
> > Table 15. Pre-training with different tasks with annotations on Waymo Dataset and fine-tune the pre-trained backbones for 3D object detection on NuScenes dataset.
> >
> > | Pre-training Task  | P.D.A. | Downstream Task | mAP | NDS |
> > | --- | --- | --- | --- | --- |
> > | From Scratch | - | 5% NuScenes | 42.37 | 52.01 |
> > | Detection Task on Upstream | 100% | 5% NuScenes | 40.89 | 49.75 |
> > | Segmentation Task on Upstream | 100% | 5% NuScenes | 36.23 | 47.01 |
> > | SPOT | 5%  | 5% NuScenes | 43.56 | 53.04 |
> > | SPOT (SEMI) | 5%  + Unlabeled 15% | 5% NuScenes | 45.18 | 54.98 |
> >
> > **Q3: Supervised occupancy prediction is not new, and It is hard to argue scalability when pre-training is based on a labeled dataset, and self-supervised occupancy pretrainig was demonstrated in previous works.**
> >
> > **A3:**  We agree with the reviewer that, supervised occupancy prediction is indeed not a new emerging field, but we are the first to build a unified pre-trained model by treating it as a pre-training task and directly adapt it to different downstream tasks and datasets. In contrast, current work such as Occupancy-MAE [1] can only be fine-tuned on the current dataset by pre-training it on the current dataset, which means that for each downstream dataset, pre-training needs to be performed again. Such a one-to-one experimental setting (perform the pre-training and fine-tuning on the same dataset) causes that the pre-trained backbone is hard to be deployed to different tasks and scenes.
> >
> > **Q4: Another critical limitation on scalability is that the pre-training dataset must have a more expensive lidar setup (64-beam with high density) compared to the fine-tuning dataset.**
> >
> > **A4:**   Building occupancy labels does not require a very expensive lidar setup, as NuScenes only has 16beam compared to Waymo's 64-beam, while the NuScenes dataset also has dense occupancy labels. Furthermore, our semi-supervised experiments also demonstrate that SPOT's reliance on occupancy labels is secondary, and more pre-training labels can be built using simple pseudo-labeling and consistent gain across all downstream tasks. Finally, we conducted pre-training experiments on the Waymo dataset with only 16-beam as input, and cross-domain experiments on the NuScenes dataset, and the experimental results show that SPOT can still achieve a good performance improvement.
> >
> > Table 16. Pre-training on the Waymo dataset with only 16beam as input and fine-tune the pre-trained backbones for 3D object detection on NuScenes dataset.
> >
> > | Method | P.D.A. | F.D.A.  | mAP | NDS |
> > | --- | --- | --- | --- | --- |
> > | From Scratch (CenterPoint) |  -  | 5% NuScenes | 42.37 | 52.01 |
> > | SPOT_5% (CenterPoint) | random 5% | 5% NuScenes | 44.42   | 54.42 |
> > | SPOT_5%_16beam (CenterPoint) | random 5% | 5% NuScenes  | 44.24   | 53.45 |

---

> ### Author Response · Authors · 2023-11-15
> **Response to Reviewer 4p2k (Part3)**
>
> **Q5: Low train from scratch baseline performance: As shown in Fig 7, with 100% NuScenes, training from scratch leads to mAP ~ 50, which is far from SOTA (>70).**
>
> **A5:** Thanks for pointing this out. Our motivation is to improve label-efficiency performance on various downstream tasks and datasets via **one** pre-trained model. We totally agree that it is also important to improve SOTA models. Most models that achieve >70 NDS performance on NuScenes Leaderboard is multi-modality methods with camera inputs but we focus on LiDAR perception. To demonstrate SPOT can improve SOTA model, we perform fine-tuning experiments with 100% full data on the VoxelNext [2] model, which is one of SOTA LiDAR-only models on NuScenes benchmark. And the results in Table 17 prove that SPOT can still achieve gains on SOTA models.
>
> Table 17. Fine-tune the pre-trained backbones for VoxelNext on 100% NuScenes 3D object detection dataset.
>
> | Method | P.D.A. | F.D.A.  | mAP | NDS |
> | --- | --- | --- | --- | --- |
> | From Scratch (VoxelNext) |  -  | 100% NuScenes | 60.53 | 66.65 |
> | SPOT  (VoxelNext) | 100% | 100% NuScenes | 61.05 | 66.90 |
>
> **Q6: Could the baseline model be undertrained or that it is using a weaker data augmentation?**
>
> **A6:**  The data augmentations we used all follow the OpenPCDet settings, including gt sampling, random flip/rotate/translate, etc., which is a consistent approach for current 3D detection models. Besides, we conduct experiments on extending the training schedule on NuScenes dataset, and the results are shown as follows. From the results, it can be seen that even when the training schedule is extending by 5 times at least, the initialization by our pre-trained model can achieve higher performance. Especially for the Transformer-based DSVT detector, the result of just 20 epochs is even superior to the result of 150 epochs.
>
> Table 18. Extending training schedule on NuScenes dataset.
>
> | Method | P.D.A. | F.D.A.  | Training Schedule | mAP | NDS |
> | --- | --- | --- | --- | --- | --- |
> | From Scratch (SECOND) |  -  | 5% NuScenes | 30 epochs | 32.16  | 41.59 |
> | From Scratch (SECOND) |  -  | 5% NuScenes | 150 epochs | 36.79  | 51.01 |
> | SPOT (SECOND) | 20% | 5% NuScenes | 30 epochs | 39.63 | 51.63 |
> | SPOT (SECOND) | 100% | 5% NuScenes | 30 epochs | 42.57  | 54.28 |
> | From Scratch (CenterPoint) | - | 5% NuScenes | 30 epochs | 42.37 | 52.01 |
> | From Scratch (CenterPoint) | - | 5% NuScenes  | 150 epochs | 41.01  | 53.92 |
> | SPOT (CenterPoint) | 20% | 5% NuScenes | 30 epochs | 44.94 | 54.95 |
> | SPOT (CenterPoint) | 100% | 5% NuScenes | 30 epochs | 47.47  | 57.11 |
> | From Scratch (DSVT) |  -  | 5% NuScenes | 20 epochs | 49.78 | 58.63 |
> | From Scratch (DSVT) |  -  | 5% NuScenes | 150 epochs | 54.30 | 63.58 |
> | SPOT (DSVT) | 20% | 5% NuScenes | 20 epochs | 56.65 | 63.52 |
>
> We hope our additional experiment results and discussion address your concerns. And we are glad to discuss anything unclear about our paper.
>
> Best regards,
>
> Authors of Paper 496
>
> [1] Chen Min, Liang Xiao, Dawei Zhao, Yiming Nie, and Bin Dai. Occupancy-mae: Self-supervised pre-training large-scale lidar point clouds with masked occupancy autoencoders. IEEE Transactions on Intelligent Vehicles, 2023.
>
> [2] Chen Y, Liu J H, Zhang X Y, et al. VoxelNeXt: Fully Sparse VoxelNet for 3D Object Detection and Tracking. arXiv 2023[J]. arXiv preprint arXiv:2303.11301.

---

> ### Author Response · Authors · 2023-11-17
> **Response to Updated Review (Part 1)**
>
> Dear Reviewer ****4p2k,****
>
> Thank you for the prompt reply and the acknowledgement of our additional experiments and discussions. We are writing to provide further discussion on your remaining concerns.
>
> **Q1: The seemingly marginal improvement for VoxelNext.**
>
> First of all, in our pre-training, both 3D and 2D (BEV) backbones are used for occupancy prediction but VoxelNext [1] only uses 3D backbone for encoding. Thus, we are only able to load weights of 3D backbone when fine-tuning on NuScenes, which losses useful information from pre-training. Under such circumstances, we also achieve 0.52\% mAP improvement. It is a well-known phenomenon in previous works [2,3,4] that improving over SOTA models with 100% downstream labels is hard and 0.52\% mAP is non-negligible improvement.
>
> To further demonstrate the ability of SPOT to improve SOTA performance on NuScenes, we perform two additional experiments: (1) training VoxelNext on train and validation set and then test it on **Test** set. (2) fine-tuning on CenterPoint [5], which is another SOTA model on NuScenes, and it has both 3D and 2D backbone for embedding point clouds, leading to less information loss.
>
> (1) We train VoxelNext on train and validation set and then test it on **Test** set, which is a standard setting for **test set**. It can be seen from the following table that, although only pre-trained parameters of 3D backbone are loaded, SPOT can boost the performance of VoxelNext on **test set** by 1.0\% mAP and 0.7\% NDS, respectively. More than 200\% improvement are gained compared to previous results and this improvement is significant on SOTA models as stated in previous works [2,3,4].
>
> Table. Fine-tune the pre-trained backbones for VoxelNext on 100% NuScenes 3D object detection trainval set and **evaluate on test set.**
>
> | Method | F.D.A.  | mAP | NDS |
> | --- | --- | --- | --- |
> | From Scratch (VoxelNext) | 100% NuScenes trainval Set | 64.5 | 70.0 |
> | SPOT  (VoxelNext) | 100% NuScenes trainval Set | **65.5** | **70.7** |
>
> (2) Furthermore, to avoid information loss, we employ the CenterPoint [5] (voxel_size=0.075), which utilizes both 3D and 2D backbones for encoding, for 3D object detection task on NuScenes. We load pre-trained weights for both 3D and 2D backbones from SPOT and fine-tune CenterPoint with full training data. In the Table below, it can be found that compared to training from scratch, initializing CenterPoint by SPOT brings 2.63\% mAPs and 1.25\% NDS improvement. Thus, without loss of pre-trained information, that is loading the full amount of pre-training parameters, SPOT is able to achieve far more significant performance improvement on SOTA models.
>
> Table. Fine-tune the pre-trained backbones for CenterPoint (voxel_size=0.075) on 100% NuScenes 3D object detection dataset.
>
> | Method | F.D.A.  | mAP | NDS |
> | --- | --- | --- | --- |
> | From Scratch (CenterPoint) | 100% NuScenes train set | 59.28 | 66.60 |
> | SPOT  (CenterPoint) | 100% NuScenes train set | **61.91** | **67.85** |
>
> We hope these discussions and experiment results can cover your concerns on the improvement on SOTA models. Thanks.
>
> ---
>
> ---

---

> > ### Author Response · Authors · 2023-11-17
> > **Response to Updated Review (Part 2)**
> >
> > **Q2:** As reviewer y7L9 suggested, `the improvements and results showcased appear anticipated`.
> >
> > We provide discussion on this concern in **Q2** of [Response to Reviewer y7L9](https://openreview.net/forum?id=9zEBK3E9bX&noteId=ctrbWDv7Xw) and we are glad to provide a further discussion here on why we believe that purely using large-scale annotated dataset for pre-training to bring the improvements is **not anticipated.**
> >
> > Firstly, it has been demonstrated in previous works [6,7] that directly using Waymo annotated data ****cannot**** help improve the performance of other datasets such as NuScenes, which is mainly due to huge domain differences between different 3D datasets,  especially for the NuScenes dataset. Also, the works [8,9] also shows that the pre-trained model on the Waymo dataset has very limited performance when transferring to other datasets. Cross-domain adaptation between different datasets is still one of the challenges in the field of 3D outdoor point cloud perception. We cite the results reported from [6] here. From the following table, it can also be observed that, large-scale annotated Waymo dataset does not improve the model performance on NuScenes dataset, but brings ****a significant performance drop**** on NuScenes. The performance drop mainly is due to the domain differences (including class definitions, sensor setting and scenarios differences) from both datasets.
> >
> > Table. Domain adaptation performance under Waymo→NuScenes with different amounts of target-domain labels between Waymo and NuScenes. **Note that the results here are evaluated and reported only in the car category [6], which is different from the overall performance reported in our experiments.**
> >
> > | Training Method | 5% NuScenes (mAP/ NDS on the category of CAR) | 10% NuScenes (mAP/ NDS on the category of CAR) | 20% NuScenes (mAP/ NDS on the category of CAR) | 100% NuScenes (mAP/ NDS on the category of CAR) |
> > | --- | --- | --- | --- | --- |
> > | Only NuScenes | 61.0 / 53.2 | 65.6 / 58.2 | 70.2 / 63.0 | 78.4 / 69.9 |
> > | Using extra large-scale Waymo annotated data | 57.7 / 58.0 | 59.4 / 58.9 | 63.1 / 61.1 | 66.5 / 63.5 |
> >
> > Secondly, SPOT demonstrates the superiority of occupancy prediction as pre-training tasks to learn general 3D representations. By “general”, we mean that SPOT only pre-train the backbones for one time and these pre-trained weights are able to improve different baselines on various downstream datasets and tasks. In comparison, pre-training by 3D object detection or LiDAR semantic segmentation tasks with fully-annotated Waymo datasets suffers from two gaps when fine-tuned on downstream tasks: (1) The gap between pre-training task and downstream task. In the Table below, it can be found that pre-training on semantic segmentation task harms the performance a lot. (2) The gap between different LiDARs used to collect datasets. When pre-training via detection task, the performance on downstream dataset also drops due to different sensors used in these two datasets. Thus **annotation itself does not directly bring improvements**.
> >
> > Table. Pre-training with different tasks with annotations on Waymo Dataset and fine-tune the pre-trained backbones for 3D object detection on NuScenes dataset.
> >
> > | Pre-training Task  | P.D.A. | Downstream Task | mAP | NDS |
> > | --- | --- | --- | --- | --- |
> > | From Scratch | - | 5% NuScenes | 42.37 | 52.01 |
> > | Detection Task on Upstream | 100% | 5% NuScenes | 40.89 | 49.75 |
> > | Segmentation Task on Upstream | 100% | 5% NuScenes | 36.23 | 47.01 |
> > | SPOT | 5%  | 5% NuScenes | 43.56 | 53.04 |
> > | SPOT (SEMI) | 5%  + Unlabeled 15% | 5% NuScenes | 45.18 | 54.98 |
> >
> > Thirdly, purely applying occupancy prediction for pre-training **is not enough.** Our proposed beam-sampling augmentation and class-balancing strategies also play important role in achieving such general 3D representations. As shown in the Table below, we conduct ablation study on different components of SPOT  and it can be found that purely applying occupancy prediction as pre-training task bring **much less improvement** than that of SPOT.
> >
> > Table. Ablation study on pre-training strategies and fine-tuning performance across different datasets.
> >
> > | Occupancy Prediction | Loss Balancing   | Beam Re-sampling | Dataset Balancing | NuScenes (mAP / NDS) | ONCE(mAP) | KITTI(mAP) |
> > | --- | --- | --- | --- | --- | --- | --- |
> > |  |  |  |  | 32.16 / 41.59  | 35.96  | 61.70 |
> > | √ |  |  |  | 36.55 / 46.98  | 36.00  | 63.70 |
> > | √ | √ |  |  | 37.90 / 47.82  | 37.30  | 64.70 |
> > | √ | √ | √ |  | 38.63 / 48.85  | 39.19  | 65.92 |
> > | √ | √ | √ | √ | 40.39 / 51.65 | 40.63  | 66.45 |

---

> > > ### Author Response · Authors · 2023-11-17
> > > **Response to Updated Review (Part 3)**
> > >
> > > Thus, we think that the improvement after using large-scale labeled dataset for pre-training is **not anticipated.** SPOT actually brings new observations and insights for LiDAR perception pre-training. And together with the semi-supervised pre-training experiment with SPOT in [General Response 2](https://openreview.net/forum?id=9zEBK3E9bX&noteId=mx7XgdJGbX), we believe that SPOT would pave the way for future large scale LiDAR pre-training.
> > >
> > > We hope these discussions and results can cover this concern.
> > >
> > > We are glad to have any further discussions. Look forward to your reply!
> > >
> > > Best regards,
> > >
> > > Authors of Paper 496
> > >
> > > References:
> > >
> > > [1] Chen Y, Liu J H, Zhang X Y, et al. VoxelNeXt: Fully Sparse VoxelNet for 3D Object Detection and Tracking. arXiv 2023[J]. arXiv preprint arXiv:2303.11301.
> > >
> > > [2] Jiakang Yuan, Bo Zhang, Xiangchao Yan, Tao Chen, Botian Shi, Yikang Li, and Yu Qiao. Ad-pt: Autonomous driving pre-training with large-scale point cloud dataset. arXiv preprint arXiv:2306.00612, 2023
> > >
> > > [3] Runsen Xu, Tai Wang, Wenwei Zhang, Runjian Chen, Jinkun Cao, Jiangmiao Pang, and Dahua
> > > Lin. Mv-jar: Masked voxel jigsaw and reconstruction for lidar-based self-supervised pre-training.
> > > In Proceedings of the IEEE/CVF Conference on Computer Vision and Pattern Recognition, pp.
> > > 13445–13454, 2023.
> > >
> > > [4] Runjian Chen, Yao Mu, Runsen Xu, Wenqi Shao, Chenhan Jiang, Hang Xu, Zhenguo Li, and Ping Luo. Coˆ 3: Cooperative unsupervised 3d representation learning for autonomous driving. arXiv preprint arXiv:2206.04028, 2022.
> > >
> > > [5] Yin, Tianwei, Xingyi Zhou, and Philipp Krahenbuhl. "Center-based 3d object detection and tracking." *Proceedings of the IEEE/CVF conference on computer vision and pattern recognition*. 2021.
> > >
> > > [6] Yan Wang, Junbo Yin, Wei Li, Pascal Frossard, Ruigang Yang, and Jianbing Shen. Ssda3d: Semi-supervised domain adaptation for 3d object detection from point cloud. In Proceedings of the AAAI Conference on Artificial Intelligence, volume 37, pp. 2707–2715, 2023b.
> > >
> > > [7] Bo Zhang, Jiakang Yuan, Botian Shi, Tao Chen, Yikang Li, and Yu Qiao. Uni3d: A unified baseline for multi-dataset 3d object detection. In Proceedings of the IEEE/CVF Conference on Computer Vision and Pattern Recognition, pp. 9253–9262, 2023.
> > >
> > > [8] Jihan Yang, Shaoshuai Shi, Zhe Wang, Hongsheng Li, and Xiaojuan Qi. St3d: Self-training for unsupervised domain adaptation on 3d object detection. In Proceedings of the IEEE/CVF Conference on Computer Vision and Pattern Recognition, pp. 10368–10378, 2021.
> > >
> > > [9] Jiakang Yuan, Bo Zhang, Xiangchao Yan, Tao Chen, Botian Shi, Yikang Li, and Yu Qiao. Bi3d: Bi-domain active learning for cross-domain 3d object detection. In Proceedings of the IEEE/CVF Conference on Computer Vision and Pattern Recognition, pp. 15599–15608, 2023b.

---

> ### Author Response · Authors · 2023-11-21
> **Look forward to further discussion.**
>
> Dear Reviewer 4p2k,
>
> We hope that our response can address your concerns. As the deadline for discussion is approaching, we would greatly appreciate it if you could let us know if there are any other questions or issues regarding the paper or our response. We are looking forward to further discussion.
>
> Best regards,
>
> Authors of Paper 496

---

> ### Author Response · Authors · 2023-11-22
> **Look forward to further discussion.**
>
> Dear Reviewer 4p2k,
>
> Thank you for your precious time on the review. As the deadline of discussion period is approaching, we sincerely hope that our response can address your concerns and we are looking forward to further discussion on any other questions or issues regarding the paper or our response.
>
> Best regards,
>
> Authors of Paper 496

---

> ### Author Response · Authors · 2023-11-23
> **Sincere Request for Further Discussions**
>
> Dear Reviewer 4p2k,
>
> Thanks again for your great efforts and constructive advice in reviewing this paper! With the discussion period drawing to a close, we expect your feedback and thoughts on our reply. We put a significant effort into our response, with several new experiments and discussions. We sincerely hope you can consider our reply in your assessment. We look forward to hearing from you, and we can further address unclear explanations and remaining concerns if any.
>
> Best regards,
>
> Authors of Paper 496

---

### Official Review · Reviewer_y7L9 · 2023-10-30

**Soundness:** 2 fair
**Presentation:** 3 good
**Contribution:** 2 fair
**Rating:** 3
**Confidence:** 4

**Summary:**

The paper presents SPOT (Scalable Pre-training via Occupancy Prediction), aimed at learning transferable 3D representations from LiDAR point clouds for autonomous driving tasks. SPOT leverages occupancy prediction as a pre-training task to learn general representations, employs beam re-sampling for point cloud augmentation, and class-balancing strategies to bridge domain gaps caused by varying LiDAR sensors and annotation strategies across different datasets. The authors extensively test SPOT across multiple datasets and 3D perception tasks, demonstrating its scalability and effectiveness.

**Strengths:**

1. Leveraging occupancy perception as a pretraining task is interesting.
2. The tricks of beam re-sampling augmentation and class-balancing strategies are useful.
3. The authors did large-scale experiments on five datasets.

**Weaknesses:**

My major concern is that this pretraining is not self-supervised representation learning. Although the authors tout SPOT as a label-efficient solution, the labor-intensive nature of building a large-scale semantic occupancy dataset seems to contradict this claim. Furthermore, the improvements and results showcased appear anticipated, especially given the employment of the extensively annotated large-scale Waymo open dataset.

**Questions:**

What will the results look like if only using binary occupancy labels without any human labeling? How can the pretraining be made fully self-supervised? Meanwhile, what will the results look like if pretraining on a different dataset (not Waymo)?

---

> ### Author Response · Authors · 2023-11-15
> **Response to Reviewer y7L9 (Part 1)**
>
> Dear Reviewer y7L9,
>
> Thank you for your review. We conduct new experiments on semi-supervised pre-training and provide detailed analysis both in [General Response Part 2](https://openreview.net/forum?id=9zEBK3E9bX&noteId=mx7XgdJGbX) and here. We also provide experiment results on pre-training on other tasks and pre-training on NuScenes dataset via SPOT to cover your concerns. Besides, we provide discussion on supervised, semi-supervised and self-supervised pre-training here.
>
> **Q1: Label-intensive nature for pre-training via occupancy prediction.**
>
> **A1:** We thank you for acknowledging SPOT as a promising pre-training method for LiDAR perception and pointing out this aspect. We totally agree that using fewer or no labels for pre-training, namely semi-supervised or self-supervised pre-training, is essential and more practical. To address this concern, we conduct semi-supervised pre-training with SPOT. To be more specific, we pre-train the backbones with SPOT using only 5% sequence-level labeled data and different amount of sequence-level unlabeled data (5% and 15%) via pseudo-labeling [1] and mean-teacher approach [2]. The results are shown in Tables 1, 2, 3, 4 and 5 in General Response Part 2.
>
> It can be found in Tables 1, 2, 3, 4 and 5 that pre-training with 5% sequence-level labeled data and 15% sequence-level unlabeled data achieves comparable downstream fine-tuning performance as that of pre-training with 20% sequence-level labeled data. Meanwhile, when compared to pre-training with 5% sequence-level labeled data and 5% sequence-level unlabeled data, it can be found that **more unlabeled data brings better downstream performance**. These results verify that (1) SPOT can learn general representations for different downstream datasets, tasks and architectures with small part of labels. (2) When incorporating more unlabeled data into pre-training phase, SPOT also shows scalability.
>
> We hope these results can cover your concerns on the label-intensive nature for fully-supervised pre-training via SPOT.
>
> **Q2: Given annotations on Waymo Dataset, the improvements and results are anticipated**
>
> **A2:** We would like to clarify that the proposed SPOT is **a more general pre-training paradigm**. This means that only one pre-trained is conducted via SPOT, and we applied the pre-trained backbones to improve performance in various datasets and tasks. As shown in Table 8 below, it can be found that pre-training via detection or segmentation task on fully-annotated Waymo data suffers from two gaps when fine-tuned in downstream tasks (1) the gap between pre-training task and downstream task. Pre-training on semantic segmentation task harms the performance a lot. (2) the gap between different LiDARs used to collect datasets. When pre-training via detection task, the performance on downstream dataset also drops due to different sensors used in these two datasets. In comparison, SPOT achieves general improvements, which demonstrate the superiority of SPOT as pre-training task.
>
> Table 8. Pre-training with different tasks with annotations on Waymo Dataset and fine-tune the pre-trained backbones for 3D object detection on NuScenes dataset.
>
> | Pre-training Task  | P.D.A. | Downstream Task | mAP | NDS |
> | --- | --- | --- | --- | --- |
> | From Scratch | - | 5% NuScenes | 42.37 | 52.01 |
> | Detection Task on Upstream | 100% | 5% NuScenes | 40.89 | 49.75 |
> | Segmentation Task on Upstream | 100% | 5% NuScenes | 36.23 | 47.01 |
> | SPOT | 5%  | 5% NuScenes | 43.56 | 53.04 |
> | SPOT (SEMI) | 5%  + Unlabeled 15% | 5% NuScenes | 45.18 | 54.98 |
>
> **Q3: Using only binary occupancy labels for pre-training**
>
> **A3:** We conduct experiments where we pre-train backbones via SPOT with only binary occupancy labels. The results are as shown below:
>
> Table 9: Pre-training via SPOT with binary labels indicating Occupied and Empty.
>
> | Method | P.D.A. | F.D.A.  | mAP | NDS |
> | --- | --- | --- | --- | --- |
> | From Scratch (CenterPoint) |  -  | 5% NuScenes | 42.37 | 52.01 |
> | Binary occupancy pre-training (CenterPoint) | 20% | 5% NuScenes | 42.05 | 51.63 |
> | SPOT  (CenterPoint) | 20% | 5% NuScenes  | 44.94  | 54.95 |
>
> It can be found that pre-training only with binary labels harm the performance. This might stem from that pure occupancy information provides little semantic information and thus offer little help for downstream detection and semantic segmentation tasks.

---

> ### Author Response · Authors · 2023-11-15
> **Response to Reviewer y7L9 (Part 2)**
>
> **Q4: How to make pre-training self-supervised via SPOT?**
>
> **A4:** We totally agree that using fewer or no labels for pre-training, namely semi-supervised or self-supervised pre-training, is essential and more practical. However, previous studies [3,4,5] that employ the fully self-supervised paradigm suffer from the inability to attain the task or dataset scalability: they perform the 3D pre-training and downstream on the same dataset. In contrast, SPOT is the first to explore the so-called one-for-all setting, meaning that the baseline model is only pre-trained once using the proposed SPOT and fine-tuned on different downstream datasets and tasks.
>
> Also, as demonstrated in image domain, where the most popular pre-training model is fully-supervised pre-trained on ImageNet [6], it is not necessary for pre-training methods to be full self-supervised. Besides, for 3D pre-training on LiDAR point cloud, AD-PT [7] is a successful exploration on semi-supervised pre-training. We perform semi-supervised pre-training experiments with SPOT in [**Q1**](https://openreview.net/forum?id=9zEBK3E9bX&noteId=ctrbWDv7Xw) and [General Response Part 2](https://openreview.net/forum?id=9zEBK3E9bX&noteId=mx7XgdJGbX), which demonstrates the effectiveness of SPOT with few labeled data.
>
> We hope these can cover your concerns.
>
> **Q5: Pre-training on a different dataset via SPOT?**
>
> **A5:** To verify that SPOT is able to pre-train on other datasets, we utilize the model which is pre-trained on Waymo to predict occupancy labels on 5% NuScenes dataset and generate pseudo occupancy labels. Next, we pre-train SPOT **from scratch on these NuScenes data**, and then fine-tune on the 20% KITTI data. As shown in Table 10 below, SPOT achieves significant gains compared to baseline results on KITTI dataset, demonstrating the effectiveness and generalization of SPOT.
>
> Table 10: Pre-training via SPOT on NuScenes and downstream to 3D object detection on KITTI.
>
> | Method | Pre-training dataset | F.D.A.  | mAP |
> | --- | --- | --- | --- |
> | From Scratch (SECOND) | - | 20% KITTI | 61.70 |
> | SPOT  (SECOND) | NuScenes | 20% KITTI | 64.39 |
> | From Scratch (PV-RCNN) | - | 20% KITTI | 66.71 |
> | SPOT  (PV-RCNN) | NuScenes | 20% KITTI | 69.58 |
>
> We hope the discussion and experiments address your concerns. We are happy to discuss any remaining concerns about the paper.
>
> Best regards,
>
> Authors of Paper 496
>
> [1] Dong-Hyun Lee et al. Pseudo-label: The simple and efficient semi-supervised learning method for deep neural networks. In Workshop on challenges in representation learning, ICML, volume 3, pp. 896. Atlanta, 2013.
>
> [2] Antti Tarvainen and Harri Valpola. Mean teachers are better role models: Weight-averaged consistency targets improve semi-supervised deep learning results. Advances in neural information processing systems, 30, 2017.
>
> [3] Runsen Xu, Tai Wang, Wenwei Zhang, Runjian Chen, Jinkun Cao, Jiangmiao Pang, and Dahua Lin. Mv-jar: Masked voxel jigsaw and reconstruction for lidar-based self-supervised pre-training. In Proceedings of the IEEE/CVF Conference on Computer Vision and Pattern Recognition, pp. 13445–13454, 2023
>
> [4] Zhiwei Lin and Yongtao Wang. Bev-mae: Bird’s eye view masked autoencoders for outdoor point cloud pre-training. arXiv preprint arXiv:2212.05758, 2022
>
> [5] Hanxue Liang, Chenhan Jiang, Dapeng Feng, Xin Chen, Hang Xu, Xiaodan Liang, Wei Zhang, Zhenguo Li, and Luc Van Gool. Exploring geometry-aware contrast and clustering harmonization for self-supervised 3d object detection. In Proceedings of the IEEE/CVF International Conference on Computer Vision, pp. 3293–3302, 2021.
>
> [6] Deng J, Dong W, Socher R, et al. Imagenet: A large-scale hierarchical image database[C]//2009 IEEE conference on computer vision and pattern recognition. Ieee, 2009: 248-255.
>
> [7] Jiakang Yuan, Bo Zhang, Xiangchao Yan, Tao Chen, Botian Shi, Yikang Li, and Yu Qiao. Ad-pt: Autonomous driving pre-training with large-scale point cloud dataset. arXiv preprint arXiv:2306.00612, 2023

---

> > ### Author Response · Authors · 2023-11-17
> > **Look forward to further discussion.**
> >
> > Dear Reviewer y7L9,
> >
> > We hope that our response can address your concerns. As the deadline for discussion is approaching, we would greatly appreciate it if you could let us know if there are any other questions or issues regarding the paper or our response. We are looking forward to further discussion.
> >
> > Best regards,
> >
> > Authors of Paper 496

---

> ### Author Response · Authors · 2023-11-21
> **Look forward to further discussion.**
>
> Dear Reviewer y7L9,
>
> The deadline of discussion period is approaching. We hope that our response can address your concerns and we are looking forward to further discussion on any remaining question about our paper and response.
>
> Best regards,
>
> Authors of Paper 496

---

> ### Author Response · Authors · 2023-11-22
> **Look forward to further discussion.**
>
> Dear Reviewer y7L9,
>
> Thank you for your precious time on the review. As the deadline of discussion period is approaching, we sincerely hope that our response can address your concerns and we are looking forward to further discussion on any other questions or issues regarding the paper or our response.
>
> Best regards,
>
> Authors of Paper 496

---

> ### Author Response · Authors · 2023-11-23
> **Sincere Request for Further Discussions**
>
> Dear Reviewer y7L9,
>
> Thanks again for your great efforts and constructive advice in reviewing this paper! With the discussion period drawing to a close, we expect your feedback and thoughts on our reply. We put a significant effort into our response, with several new experiments and discussions. We sincerely hope you can consider our reply in your assessment. We look forward to hearing from you, and we can further address unclear explanations and remaining concerns if any.
>
> Best regards,
>
> Authors of Paper 496

---

> > ### Comment · Reviewer_y7L9 · 2023-12-05
> > **Thank you for your response.**
> >
> > I will keep my rating, as the pre-training only with binary labels can even harm the performance. Although the proposed method demonstrates some improvements, the requirement of high-cost semantic occupancy labels hinders the practical value. Meanwhile, I suggest the author change "Occupancy" to "Semantic Occupancy" in the title.

---

### Author Response · Authors · 2023-11-15
**General Response (Part 3: other additional experiment results)**

We additionally conducted experiments to pre-train transformer-based 3D backbone (DSVT [1]) with SPOT and fine-tuned the pre-trained backbone for 3D object detection task on NuScenes and ONCE datasets under label-efficiency setting. Tables 6 and 7 show the results. It can be found that SPOT is able to improve performance of different kinds of 3D backbones and achieve consistent performance improvement across different datasets. With 20% pre-training data, SPOT significantly improves DSVT by more than 8\% mAP on ONCE dataset.

Table 6: Pre-training transformer-based backbone via SPOT and fine-tune on 3D object detection on NuScenes dataset

| Method | P.D.A. | F.D.A.  | mAP | NDS |
| --- | --- | --- | --- | --- |
| From Scratch (DSVT) |  -  | 5% NuScenes | 49.78 | 58.63 |
| SPOT (DSVT) | 5% | 5% NuScenes | 55.47 | 62.17 |
| SPOT (DSVT) | 20% | 5% NuScenes | 56.65  | 63.52 |

Table 7: Pre-training transformer-based backbone via SPOT and fine-tune on 3D object detection on Once dataset

| Method | P.D.A. | F.D.A.  | mAP |
| --- | --- | --- | --- |
| From Scratch (DSVT)  |  -  | 20% ONCE | 51.52 |
| SPOT (DSVT)  | 5% | 20% ONCE | 57.81 |
| SPOT (DSVT)  | 20% | 20% ONCE | 59.78 |

[1] DSVT: Dynamic Sparse Voxel Transformer With Rotated Sets. Haiyang Wang, Chen Shi, Shaoshuai Shi, Meng Lei, Sen Wang, Di He, Bernt Schiele, Liwei Wang; Proceedings of the IEEE/CVF Conference on Computer Vision and Pattern Recognition (CVPR), 2023, pp. 13520-13529

---

### Author Response · Authors · 2023-11-15
**General Response (Part 2: semi-supervised pre-training via SPOT)**

We thank Reviewers y7L9 and 4p2k for pointing out that fully-supervised pre-training is to some extend unscalable, although SPOT is promising in pre-training for LiDAR perception tasks. We totally agree that pre-training with fewer or no labels, namely semi-supervised or self-supervised pre-training, is essential and more practical. To address this concern, we conduct semi-supervised pre-training with SPOT. To be more specific, we pre-train the backbones with SPOT using only 5% sequence-level labeled data and 15% sequence-level unlabeled data via naive pseudo-labeling [1] and mean-teacher approach [2]. The results are shown in Tables 1, 2, 3, 4 and 5 below. P.D.A. is abstract for pre-training sequence-level data amount and F.D.A stands for fine-tuning data amount. L means labeled data amount and U means unlabeled data amount.

Table 1: Pre-training with 5% labeled data and different amount of unlabeled data (5% and 15%) via SPOT and fine-tune on 3D object detection on Once dataset

| Backbone    | Method | P.D.A. | F.D.A.  | mAP  |
| --- | --- | --- | --- | --- |
| SECOND | From Scratch |  -  | 20% ONCE | 35.96 |
| SECOND | SPOT | L5% | 20% ONCE | 37.98 |
| SECOND | SPOT | L20% | 20% ONCE | 39.33 |
| SECOND | SPOT (SEMI) | L5%+U5% | 20% ONCE | 38.31 |
| SECOND | SPOT (SEMI) | L5%+U15% | 20% ONCE | 39.85 |
| CenterPoint | From Scratch |  -  | 20% ONCE | 54.31 |
| CenterPoint | SPOT | L5% | 20% ONCE | 54.70 |
| CenterPoint | SPOT | L20% | 20% ONCE | 56.03 |
| CenterPoint | SPOT (SEMI) | L5%+U5% | 20% ONCE | 55.05 |
| CenterPoint | SPOT (SEMI) | L5%+U15% | 20% ONCE | 56.00 |

Table 2: Pre-training with 5% labeled data and different amount of unlabeled data (5% and 15%) via SPOT and fine-tune on 3D object detection on NuScenes dataset

| Backbone    | Method | P.D.A. | F.D.A.  | mAP  | NDS |
| --- | --- | --- | --- | --- | --- |
| SECOND | From Scratch |  -  | 5% Nuscenes | 32.16 | 41.59 |
| SECOND | SPOT | L5% | 5% Nuscenes | 37.96  | 48.45 |
| SECOND | SPOT | L20% | 5% Nuscenes | 39.63 | 51.63 |
| SECOND | SPOT (SEMI) | L5%+U5% | 5% Nuscenes | 38.50 | 50.03 |
| SECOND | SPOT (SEMI) | L5%+U15% | 5% Nuscenes | 39.81 | 51.51 |
| CenterPoint | From Scratch |  -  | 5% Nuscenes | 42.37  | 52.01 |
| CenterPoint | SPOT | L5% | 5% Nuscenes | 43.56  | 53.04 |
| CenterPoint | SPOT | L20% | 5% Nuscenes | 44.94  | 54.95 |
| CenterPoint | SPOT (SEMI) | L5%+U5% | 5% Nuscenes | 43.65  | 53.82 |
| CenterPoint | SPOT (SEMI) | L5%+U15% | 5% Nuscenes | 45.18  | 54.98 |

Table 3: Pre-training with 5% labeled data and different amount of unlabeled data (5% and 15%) via SPOT and fine-tune on 3D object detection on KITTI dataset

| Backbone    | Method | P.D.A. | F.D.A.  | mAP  |
| --- | --- | --- | --- | --- |
| SECOND | From Scratch |  -  | 20% KITTI | 61.70 |
| SECOND | SPOT | L5% | 20% KITTI | 63.53 |
| SECOND | SPOT | L20% | 20% KITTI | 65.45 |
| SECOND | SPOT (SEMI) | L5%+U5% | 20% KITTI | 65.18 |
| SECOND | SPOT (SEMI) | L5%+U15% | 20% KITTI | 66.45 |
| CenterPoint | From Scratch |  -  | 20% KITTI | 66.71 |
| CenterPoint | SPOT | L5% | 20% KITTI | 70.33 |
| CenterPoint | SPOT | L20% | 20% KITTI | 70.85 |
| CenterPoint | SPOT (SEMI) | L5%+U5% | 20% KITTI | 70.40 |
| CenterPoint | SPOT (SEMI) | L5%+U15% | 20% KITTI | 70.86 |

Table 4: Pre-training with 5% labeled data and different amount of unlabeled data (5% and 15%) via SPOT and fine-tune on LiDAR semantic segmentation on SemanticKITTI dataset

| Backbone    | Method | P.D.A. | F.D.A.  | mIOU |
| --- | --- | --- | --- | --- |
| Cylinder3D | From Scratch |  -  | 10% SemticKITTI   | 49.01 |
| Cylinder3D | SPOT | L5% | 10% SemticKITTI   | 52.50 |
| Cylinder3D | SPOT | L20% | 10% SemticKITTI   | 54.10 |
| Cylinder3D | SPOT (SEMI) | L5%+U5% | 10% SemticKITTI   | 53.62 |
| Cylinder3D | SPOT (SEMI) | L5%+U15% | 10% SemticKITTI   | 54.70 |

Table 5: Pre-training with 5% labeled data and different amount of unlabeled data (5% and 15%) via SPOT and fine-tune on LiDAR semantic segmentation on NuScenes dataset

| Backbone    | Method | P.D.A. | F.D.A.  | mIOU |
| --- | --- | --- | --- | --- |
| Cylinder3D | From Scratch |  -  | 5% Nuscenes | 45.85 |
| Cylinder3D | SPOT | L5% | 5% Nuscenes | 46.71 |
| Cylinder3D | SPOT | L20% | 5% Nuscenes | 47.84 |
| Cylinder3D | SPOT (SEMI) | L5%+U5% | 5% Nuscenes | 47.60 |
| Cylinder3D | SPOT (SEMI) | L5%+U15% | 5% Nuscenes | 48.84 |

---

> ### Author Response · Authors · 2023-11-15
> **Additional analysis for General Response (Part 2: semi-supervised pre-training via SPOT)**
>
> It can be found in Table 1, 2, 3, 4 and 5 that pre-training with 5% sequence-level labeled data and 15% sequence-level unlabeled data achieves comparable downstream fine-tuning performance as that of pre-training with 20% sequence-level labeled data. Also, when compared to pre-training with 5% sequence-level labeled data and 5% sequence-level unlabeled data, it can be found that more unlabeled data brings better downstream performance. These results verify that (1) SPOT can learn general representations for different downstream datasets, tasks and architectures with small part of labels. (2) When incorporating more unlabeled data into pre-training phase, SPOT also shows scalability.
>
> Thus, semi-supervised occupancy prediction is promising in LiDAR perception pre-training and we believe that SPOT would pave the way for future exploration in LiDAR pre-training.
>
> [1] Dong-Hyun Lee et al. Pseudo-label: The simple and efficient semi-supervised learning method for deep neural networks. In Workshop on challenges in representation learning, ICML, volume 3, pp. 896. Atlanta, 2013.
>
> [2] Antti Tarvainen and Harri Valpola. Mean teachers are better role models: Weight-averaged consistency targets improve semi-supervised deep learning results. Advances in neural information processing systems, 30, 2017.

---

### Author Response · Authors · 2023-11-15
**General Response (Part 1: summary of the reviews and rebuttal)**

Dear AC and reviewers,

Thank you for your precious time on the review and your constructive suggestions to improve our manuscript! We appreciate that reviewers acknowledge that pre-training for LiDAR perception tasks via occupancy prediction with SPOT is interesting and novel (Reviewer y7L9, 4p2k and FWLb), the experiments are extensive (Reviewer y7L9, 4p2k and FWLb), the proposed augmentation method and class-balancing strategy are useful (Reviewer y7L9) and the presentation in the paper is clear and easy-to-follow (Reviewer y7L9, 4p2k and FWLb).

Here is the summary about our new experiments in the rebuttal and the revision of the paper.

**New Experiments**

- Experiments for semi-supervised pre-training with SPOT to demonstrate its ability to scale up (Reviewer y7L9, 4p2k and FWLb)
- Experiments on pre-training with binary labels indicating Occupied or Empty. (Reviewer 4p2k)
- Experiments on using uniform sampling methods in downstream tasks. (Reviewer FWLb)
- Experiments on applying SPOT to pre-train backbones on NuScenes. (Reviewer y7L9)
- Experiments on applying SPOT to pre-train transformer-based backbone in DSVT [1] to show that SPOT is a general pre-training method for different 3D backbones.

**Revisions (highlighted in blue in the revised paper)**

- We add discussion about task-specific semi-supervised learning learning [3,4,5] in Related Work Part.
- We add citation for class-balancing sampling [2,3] in Method Part.
- We add a new related work, OCC-MAE [6], in the Related Work part.
- We add results of additional experiments on (1) pre-training via SPOT on NuScenes Dataset (2) semi-supervised pre-training with SPOT, (3) pre-training transformer-based backbone with SPOT (4) using uniform-sampling strategy in the downstream task (5) pre-training with binary labels (Occupied and Empty) in Appendix B.
- Based on all the existing results and findings, we provide in-depth view on why SPOT is scalable, transferrable and promising for future 3D general backbone pre-training in Appendix B.

We will first address common questions and provide additional experiment results in the General Response Part 2-3. Then we answer separate concerns from each reviewer. Although reviewers acknowledge that occupancy prediction (SPOT) is promising for LiDAR perception pre-training, the common concerns mainly come from that the labeling burden in the pre-training phase would harm the scalability of SPOT  (Reviewer y7L9 and 4p2k). We provide discussions and experiment results on semi-supervised pre-training using SPOT to demonstrate the scalability of SPOT in [General Response Part 2](https://openreview.net/forum?id=9zEBK3E9bX&noteId=mx7XgdJGbX). Other additional experiment results are provided in [General Response Part 3](https://openreview.net/forum?id=9zEBK3E9bX&noteId=o5LXKh0Gi5).

We hope that our reply, new experiments, and revision address your concern. Look forward to further discussion!

Best regards, Authors of paper 496

[1] DSVT: Dynamic Sparse Voxel Transformer With Rotated Sets. Haiyang Wang, Chen Shi, Shaoshuai Shi, Meng Lei, Sen Wang, Di He, Bernt Schiele, Liwei Wang; Proceedings of the IEEE/CVF Conference on Computer Vision and Pattern Recognition (CVPR), 2023, pp. 13520-13529

[2] Zou, Y., Yu, Z., Kumar, B. V. K., & Wang, J. (2018). Unsupervised domain adaptation for semantic segmentation via class-balanced self-training. In Proceedings of the European conference on computer vision (ECCV) (pp. 289-305).

[3] Unal, O., Dai, D., & Van Gool, L. (2022). Scribble-supervised lidar semantic segmentation. In *Proceedings of the IEEE/CVF Conference on Computer Vision and Pattern Recognition* (pp. 2697-2707).

[4] Li, L., Shum, H. P., & Breckon, T. P. (2023). Less is more: Reducing task and model complexity for 3d point cloud semantic segmentation. In *Proceedings of the IEEE/CVF Conference on Computer Vision and Pattern Recognition* (pp. 9361-9371).

[5] Kong, L., Ren, J., Pan, L., & Liu, Z. (2023). Lasermix for semi-supervised lidar semantic segmentation. In *Proceedings of the IEEE/CVF Conference on Computer Vision and Pattern Recognition* (pp. 21705-21715).

[6] Chen Min, Liang Xiao, Dawei Zhao, Yiming Nie, and Bin Dai. Occupancy-mae: Self-supervised pre-training large-scale lidar point clouds with masked occupancy autoencoders. IEEE Transactions on Intelligent Vehicles, 2023

---

### Author Response · Authors · 2023-11-15
**Summary of rebuttal phase**

Dear AC and Reviewers,

Many thanks for your valuable comments and constructive suggestions to improve the quality of our work. We conduct additional experiments, provide detailed discussion and update our manuscript to cover reviewers’ concerns. Here is a summary of what we have done in the rebuttal phase.

- We made several revisions to our original manuscript as stated in [Paper Update Part](https://openreview.net/forum?id=9zEBK3E9bX&noteId=QPOLz8XTYz).
- We conducted several new experiments to cover the concerns of the reviewers from the following aspects.
    - Semi-supervised Pre-training Setting with SPOT to demonstrate its ability to scale up.
    - Pre-training with binary labels indicating Occupied or Empty.
    - Pre-training SPOT with NuScenes dataset.
    - Applying our pre-trained backbone to Full-label setting and SOTA model in downstream task.
    - Using uniform sampling methods in downstream tasks.
    - Evaluate the effectiveness of SPOT on different 3D backbones, specifically transformer-based backbone in DSVT [1], to show the generalization ability of SPOT.

- We provide further discussion on different pre-training paradigms (including supervised, semi-supervised and self-supervised pre-training) and why SPOT is scalable, transferrable and promising for future 3D general backbone pre-training.
- Based on the existing experiment results, we provide discussions on the differences and connections between of pre-training via occupancy prediction and previous task-specific semi-supervised methods.
- Based on all the experiment results and discussion above, we provide in-depth view on why SPOT is scalable, transferrable and promising for future 3D general backbone pre-training.

Thank you again for your precious time. We hope that our response addresses your concerns and we are happy to have further discussion on anything unclear about our paper.

Best regards,

Authors of Paper 496

[1] DSVT: Dynamic Sparse Voxel Transformer With Rotated Sets. Haiyang Wang, Chen Shi, Shaoshuai Shi, Meng Lei, Sen Wang, Di He, Bernt Schiele, Liwei Wang; Proceedings of the IEEE/CVF Conference on Computer Vision and Pattern Recognition (CVPR), 2023, pp. 13520-13529

---

### Author Response · Authors · 2023-11-15
**Paper Update**

Dear AC and Reviewers,

Thank you for your precious time and constructive suggestions to improve the quality of our manuscript. We uploaded a new version of our paper and the updates made during the rebuttal period are highlighted in blue in the revision. We provide a summary as below:

* We add discussion about task-specific semi-supervised learning/active learning [1,2,3] in the Related Work part.
* We add related work on class-balancing sampling [1,4] in Method Part.
* We add a new related work, OCC-MAE [5], in the Related Work part.
* We add results of additional experiments on (1) pre-training via SPOT on NuScenes Dataset (2) **semi-supervised pre-training with SPOT**, (3) pre-training transformer-based backbone with SPOT (4) using uniform-sampling strategy in the downstream task (5) pre-training with binary labels (Occupancy and Empty) in Appendix B.
* Based on all the existing results and findings, we provide in-depth view on why SPOT is scalable, transferrable and promising for future 3D general backbone pre-training in Appendix B.

We hope that our revision makes our presentation clearer and covers reviewers' concerns. We are happy to discuss any remaining questions about our work.

Best regards,

Authors of Paper 496



[1] Unal, O., Dai, D., & Van Gool, L. (2022). Scribble-supervised lidar semantic segmentation. In *Proceedings of the IEEE/CVF Conference on Computer Vision and Pattern Recognition* (pp. 2697-2707).

[2] Li, L., Shum, H. P., & Breckon, T. P. (2023). Less is more: Reducing task and model complexity for 3d point cloud semantic segmentation. In *Proceedings of the IEEE/CVF Conference on Computer Vision and Pattern Recognition* (pp. 9361-9371).

[3] Kong, L., Ren, J., Pan, L., & Liu, Z. (2023). Lasermix for semi-supervised lidar semantic segmentation. In *Proceedings of the IEEE/CVF Conference on Computer Vision and Pattern Recognition* (pp. 21705-21715).

[4] Zou, Y., Yu, Z., Kumar, B. V. K., & Wang, J. (2018). Unsupervised domain adaptation for semantic segmentation via class-balanced self-training. In Proceedings of the European conference on computer vision (ECCV) (pp. 289-305).

[5] Chen Min, Liang Xiao, Dawei Zhao, Yiming Nie, and Bin Dai. Occupancy-mae: Self-supervised pre-training large-scale lidar point clouds with masked occupancy autoencoders. IEEE Transactions on Intelligent Vehicles, 2023

---

### Meta-Review · Area_Chair_cD8U · 2023-12-07

**Metareview:**

The paper proposes Pre-training a representation for autonomous driving tasks from LiDAR point clouds. Originally, the paper proposed occupancy prediction as pre-training objective. However, upon two reviewers who pointed out that pre-training was supervised and consequently needed expensive labels, the authors ran a series of semi-supervised experiments as well.

While I appreciate the effort the authors put into the new experiments, I am recommending a reject because the newer experiments materially change the focus of the paper - my recommendation for the authors is to resubmit with the changes to another venue.

**Justification For Why Not Higher Score:**

As stated above the major weakness was supervised nature of pre-training, which is expensive and associated experiments do not necessarily are fair on other baselines. The new experiments during the rebuttal time significantly changes the paper - hence not suited for acceptance currently.

**Justification For Why Not Lower Score:**

N/A

---

### Decision · Program_Chairs · 2024-01-16

Reject